# Bayesian calibration, process modeling and uncertainty quantification in biotechnology

**Laura Marie Helleckes**[1,2☯], **Michael Osthege**[1,2☯], **Wolfgang Wiechert**[1,3], **Eric von Lieres**[1], **Marco Oldiges**[1,2]*

**1** Institute of Bio- and Geosciences, IBG-1: Biotechnology, Forschungszentrum Jülich GmbH, Jülich, Germany, **2** Institute of Biotechnology, RWTH Aachen University, Aachen, Germany, **3** Computational Systems Biotechnology (AVT.CSB), RWTH Aachen University, Aachen, Germany

☯ These authors contributed equally to this work.
* m.oldiges@fz-juelich.de

**Data Availability Statement:** All relevant data are within the manuscript and its Supporting information files. The code of our software packages as well as detailed documentation with

## Abstract

High-throughput experimentation has revolutionized data-driven experimental sciences and opened the door to the application of machine learning techniques. Nevertheless, the quality of any data analysis strongly depends on the quality of the data and specifically the degree to which random effects in the experimental data-generating process are quantified and accounted for. Accordingly calibration, i.e. the quantitative association between observed quantities and measurement responses, is a core element of many workflows in experimental sciences.

Particularly in life sciences, univariate calibration, often involving non-linear saturation effects, must be performed to extract quantitative information from measured data. At the same time, the estimation of uncertainty is inseparably connected to quantitative experimentation. Adequate calibration models that describe not only the input/output relationship in a measurement system but also its inherent measurement noise are required. Due to its mathematical nature, statistically robust calibration modeling remains a challenge for many practitioners, at the same time being extremely beneficial for machine learning applications.

In this work, we present a bottom-up conceptual and computational approach that solves many problems of understanding and implementing non-linear, empirical calibration modeling for quantification of analytes and process modeling. The methodology is first applied to the optical measurement of biomass concentrations in a high-throughput cultivation system, then to the quantification of glucose by an automated enzymatic assay. We implemented the conceptual framework in two Python packages, `calibr8` and `murefi`, with which we demonstrate how to make uncertainty quantification for various calibration tasks more accessible. Our software packages enable more reproducible and automatable data analysis routines compared to commonly observed workflows in life sciences.

Subsequently, we combine the previously established calibration models with a hierarchical Monod-like ordinary differential equation model of microbial growth to describe multiple replicates of *Corynebacterium glutamicum* batch cultures. Key process model parameters are learned by both maximum likelihood estimation and Bayesian inference, highlighting the flexibility of the statistical and computational framework.

application examples are provided to the readership (https://github.com/JuBiotech, https://murefi.readthedocs.io, https://calibr8.readthedocs.io) to enable straightforward dissemination in the scientific community.

**Funding:** This work was funded by the German Federal Ministry of Education and Research (BMBF, https://www.bmbf.de/) as part of the project "Digitalization In Industrial Biotechnology", DigInBio (Grant No. 031B0463A) and MOld received this funding to support the PhD thesis of MO. Further funding was received from the Enabling Spaces Program "Helmholtz Innovation Labs" of the German Helmholtz Association (https://www.helmholtz.de/) to support the "Microbial Bioprocess Lab – A Helmholtz Innovation Lab" and MOld received this funding to support the PhD thesis of LH. The funders had no role in study design, data collection and analysis, decision to publish, or preparation of the manuscript.

**Competing interests:** The authors have declared that no competing interests exist.

## Author summary

In experimental fields like biotechnology, scientists need to quantify process parameters such as concentrations and state the uncertainty around them. However, measurements rarely yield the desired quantity directly; for example, the measurement of scattered light is just an indirect measure for the number of cells in a suspension. For reliable interpretation, scientists must determine the uncertainty around the underlying quantities of interest using statistical methods.

A key step in these workflows is the establishment of calibration models to describe the relation between the quantities of interest and measurement outcomes. This is typically done using measurements of reference samples for which the true quantities are known. However, implementing and applying these statistical models often requires skills that are not commonly taught.

We therefore developed two software packages, `calibr8` and `murefi`, to simplify such calibration and modeling procedures. To showcase our work, we performed an experiment commonly seen in microbiology: the acquisition of a microbial growth curve, in this case of *Corynebacterium glutamicum*, in an online measurement device. Using our software, we built a mathematical model of the overall process to quantify relevant parameters with uncertainty, e.g. the growth rate or the yield of biomass per amount of glucose.

This is a *PLOS Computational Biology* Software paper.

# 1 Introduction

## 1.1 Calibration in life sciences

Calibration modeling is an omnipresent task in experimental science. Particularly the life sciences make heavy use of calibration modeling to achieve quantitative insights from experimental data. The importance of calibration models (also known as calibration curves) in bioanalytics is underlined in dedicated guidance documents by EMA and FDA [1, 2] that also make recommendations for many related aspects such as method development and validation. While liquid chromatography and mass spectrometry are typically calibrated with linear models [3], a four- or five-parameter logistic model is often used for immuno- or ligand-binding assays [2, 4–6]. The aforementioned guidance documents focus on health-related applications, but there are countless examples where (non-linear) calibration needs to be applied across biological disciplines. From dose-response curves in toxicology to absorbance or fluorescence measurements, or the calibration of online measurement systems, experimentalists are confronted with the task of calibration.

At the same time, recent advances in affordable liquid-handling robotics facilitate lab scientists in chemistry and biotechnology to (partially) automate their specialized assays (e.g. [7, 8]). Moreover, advanced robotic platforms for parallelized experimentation, monitoring and analytics [8, 9] motivate online data analysis and calibration for process control of running experiments.

## 1.2 Generalized computational methods for calibration

Experimental challenges in calibration are often unique to a particular field and require domain knowledge to be solved. At the same time, the statistical or computational aspects of the workflow can be generalized across domains. With the increased amount of available data in high-throughput experimentation comes the need for equally rapid data analysis and calibration. As a consequence, it is highly desirable to develop an automatable, technology-agnostic and easy-to-use framework for quantitative data analysis with calibration models.

From our perspective of working at the intersection between laboratory automation and modeling, we identified a set of requirements for calibration: Data analyses rely more and more on scripting languages such as Python or R, making the use of spreadsheet programs an inconvenient bottleneck. At various levels, and in particular when non-linear calibration models are involved, the statistically sound handling of uncertainty is at the core of a quantitative data analysis.

Before going into detail about the calibration workflow, we would like to highlight its most important aspects and terminology based on the definition of calibration by the *International Bureau of Weights and Measures* (BIPM) [10]:

> **2.39 calibration**: "Operation that, under specified conditions, in a first step, establishes a relation between the quantity values with measurement uncertainties provided by measurement standards and corresponding indications with associated measurement uncertainties and, in a second step, uses this information to establish a relation for obtaining a **measurement result** from an indication."

> **2.9 measurement result**: "[. . .] A measurement result is generally expressed as a single measured quantity value and a measurement uncertainty."

The "first step" from the BIPM definition is the establishment of a relation that we will call calibration model henceforth. In statistical terminology, the relationship is established between an independent variable (BIPM: quantity values) and a dependent variable (BIPM: indications) and it is important to note that the description of measurement uncertainty is a central aspect of a calibration model. In the application ("second step") of the calibration model, the quantification of uncertainty is a core aspect as well.

Uncertainty arises from the fact that measurements are not exact, but subject to some form of random effects. While many methods such as linear regression assume that these random effects are distributed according to a Normal distribution, we want to stress that a generalized framework for calibration should not make such constraints. Instead, domain experts should be enabled to choose a probability distribution that is best suited to describe their measurement system at hand.

Going beyond the BIPM definition, we see the application of calibration models two-fold:

- Inference of individual independent quantity values from one or more observations.

- Inferring the parameters of a more comprehensive process model from measurement responses obtained from (samples of) the system.

For both applications, uncertainties should be a standard outcome of the analysis. In life sciences, the commonly used estimate of uncertainty is the confidence interval. The interpretation of confidence intervals however is challenging, as it is often oversimplified and confused with other probability measures [11, 12]. Furthermore, their correct implementation for non-linear calibration models, and particularly in combination with complex process

models, is technically demanding. For this reason, we use Bayesian credible intervals that are interpreted as the range in which an unobserved parameter lies with a certain probability [13]. In Section 2.3 we go into more details about the uncertainty measures and how they are obtained and interpreted.

Even though high-level conventions and recommendations exist, the task of calibration is approached with different statistical methodology across the experimental sciences. In laboratory automation, we see a lack of tools enabling practitioners to build tailored calibration models while maintaining a generalized approach. At the same time, generalized calibration models have the potential to improve adequacy of complex simulations in the related fields.

While numerous software packages for modeling biological systems are available, most are targeted towards complex biological networks and do not consider calibration modeling or application to large hierarchical datasets. Notable examples are Data2Dynamics [14] or PESTO [15], both allowing to customize calibration models and the way the measurement error is described. However, both tools are implemented in MATLAB and are thus incompatible with data analysis workflows that leverage the rich ecosystem of scientific Python libraries. Here, Python packages such as PyCoTools3 [16] for the popular COPASI software [17] provide valuable functionality, but are limited with respect to custom calibration models, especially in a Bayesian modeling context. To the best of our knowledge, none of these frameworks provide customizable calibration models that can be used outside of the process modeling context and are at the same time compatible with Bayesian modeling as well as modular combination with other libraries.

## 1.3 Aim of this study

This study aims to build an understanding of how calibration models can be constructed to describe both location and spread of measurement outcomes such that uncertainty can be quantified. In two parts, we demonstrate a toolbox for calibration models, `calibr8`, on the basis of application examples, thus showing how it directly addresses questions typical for quantitative data analysis.

In part one (Section 4.1) we demonstrate how to construct such calibration models based on a reparametrized asymmetric logistic function applied to a photometric assay. We give recommendations for obtaining calibration data and introduce accompanying open-source Python software that implements object-oriented calibration models with a variety of convenience functions.

In part two (Section 4.2) we show how calibration models can become part of elaborate process models to accurately describe measurement uncertainty caused by experimental limitations. We introduce a generic framework, `murefi`, for refining a template process model into a hierarchical model that flexibly shares parameters across experimental replicates and connects the model prediction with observed data via the previously introduced calibration models. This generic framework is applied to build an ordinary differential equation (ODE) process model for 28 microbial growth curves gained in automated, high-throughput experiments. Finally, we demonstrate how the calibration model can be applied to perform maximum likelihood estimation (MLE) or Bayesian inference of process model parameters while accounting for non-linearities in the experimental observation process.

Although this paper chooses biotechnological applications, the presented approach is generic and our Python implementations are applicable to a wide range of research fields. Our documentation includes examples from broader life-science applications such as cell-counting and enzyme catalysis, which can be transferred to statistically similar problems in, for example, environmental research or chemistry.

## 2 Theoretical background

### 2.1 Probability theory for calibration modeling

Probability distributions are at the heart of virtually all statistical and modeling methods. They describe the range of values that a variable of unknown value, also called random variable, may take, together with how likely these values are. This work focuses on univariate calibration tasks, where a continuous variable is obtained as the result of the measurement procedure. Univariate, continuous probability distributions such as the Normal or Student-$t$ distribution are therefore relevant in this context. Probability distributions are described by a set of parameters, such as $\{\mu, \sigma\}$ in the case of a Normal distribution, or $\{\mu, scale, \nu\}$ in the case of a Student-$t$ distribution.

To write that a random variable "rv" follows a certain distribution, the $\sim$ symbol is used: rv $\sim$ Student-$t(\mu, scale, \nu)$. The most commonly found visual representation of a continuous probability distribution is in terms of its probability density function (PDF, S1 Fig), typically written as $p(\text{rv})$.

The term *rv conditioned on d* is used to refer to the conditional probability of a random variable rv given that certain data d was observed. It is written as $p(\text{rv} \mid \text{d})$.

A related term, the likelihood $\mathcal{L}$, takes the inverse perspective on how likely it is to make observations d given a fixed value of the random variable. Both $p(\text{d} \mid \text{rv})$ and $\mathcal{L}(\text{rv} \mid \text{d})$ are common notations for the likelihood. Note that the likelihood is not a PDF of the random variable [18]. We use the notation of $\mathcal{L}$ throughout this paper for better visibility of likelihoods.

In situations where only limited data is available, a Bayesian statistician argues that prior information should be taken into account. The likelihood can then be combined with prior beliefs into the posterior probability according to Bayes' rule (Eq 1).

$$p_{\text{posterior}}(\text{rv}|\text{d}) = \frac{p_{\text{prior}}(\text{rv}) \cdot \mathcal{L}(\text{rv}|\text{d})}{\int p_{\text{prior}}(\text{rv}) \cdot \mathcal{L}(\text{rv}|\text{d}) \; d\,\text{rv}} \tag{1}$$

According to Eq 1, the posterior probability $p(\text{rv}|\text{d})$ of the random variable rv given the data is equal to the product of prior probability times likelihood, divided by its integral.

When only considering the observed data, the probability of the random variable conditioned on data $p(\text{rv} \mid \text{d})$, can be obtained by normalizing the likelihood by its integral (Eq 2).

$$p_{\text{likelihood}}(\text{rv}|\text{d}) = \frac{\mathcal{L}(\text{rv}|\text{d})}{\int \mathcal{L}(\text{rv}|\text{d}) \; d\,\text{rv}} \tag{2}$$

From the Bayesian perspective, Eq 2 can be understood as a special case of Bayes' rule (Eq 1) with flat (uninformative) prior information. This connection between a likelihoodist and a Bayesian perspective on independent variable probabilities is illustrated in Fig 1. In Fig 1A, the red probability density function was obtained only from the likelihood of observations (black arrows), corresponding to Eq 2 or the Bayesian perspective with a flat prior (blue). Assuming that the independent variable is *a priori* known to be $\geq 1$, one might choose a corresponding prior (Fig 1B, blue). The posterior (orange) then compromises prior and likelihood according to Eq 1, resulting in a posterior probability of 0 that the variable is below 1. For a thorough introduction on Bayesian methods, we refer the interested reader to [19].

### 2.2 Parameter estimation

A mathematical model $\phi$ is a function that describes the state of system variables by means of a set of parameters. The model is a representation of the underlying data generating process, meaning that the model output from a given set of parameters is imitating the expected output

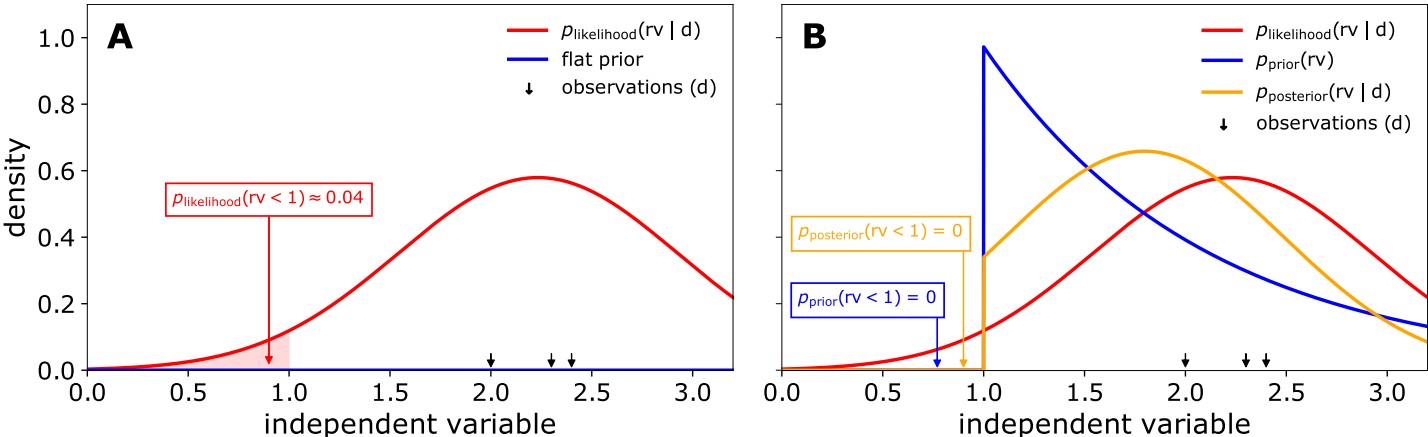

**Fig 1. Comparison of likelihoodist and Bayesian probability densities.** Using a $\mathcal{N}$ noise model, the likelihood function normalized by its integral gives a likelihoodist probability density function (red) for the independent variable. In **A** the resulting probability that the variable of interest lies below 1 is clearly positive. In a hypothetical scenario where the variable of interest is known to be $\geq 1$, a shifted exponential prior could be assigned (**B**). The posterior probability (orange) is then obtained via Bayes' rule (Eq 1) and gives the desired 0 probability of the variable being less than 1. **A** may be viewed as a special case of the Bayesian perspective with a flat prior (blue). Observations and distribution parameters were chosen to obtain a good layout: $\sigma = 1.2$, $\frac{1}{\lambda} = 1.1$, d = [2.0, 2.3, 2.4].

in the real system. From a given list of parameters $\vec{\theta}$, a model can make predictions of the system variables, in the following denominated as $\vec{y}_{\mathrm{pred}}$. In machine learning, this quantity is often called $\hat{\vec{y}}$.

$$\vec{y}_{\mathrm{pred}} = \phi(\vec{\theta}) \tag{3}$$

A predictive model can be obtained when the parameters are estimated from observed experimental data $\vec{y}_{\mathrm{obs}}$. In this process, the experimental data is compared to data predicted by the model. In order to find the prediction matching the data best, different approaches of parameter estimation can be applied. This process is sometimes also referred to as inference or informally as fitting.

To obtain one parameter vector, optimization of so-called loss functions or objective functions can be applied. In principle, these functions compare prediction and measurement outcome, yielding a scalar that can be minimized. Various loss functions such as the mean absolute error (MAE or L1 loss) or the mean squared error (MSE or L2 loss) can be formulated for the optimization process.

In the following, we first consider a special case, least squares estimation using the MSE, before coming to the generalized approach of maximum likelihood estimation. The former, which is often applied in biotechnology in the context of linear regression, is specified in the following equation:

$$L = \left(\vec{y}_{\mathrm{obs}} - \vec{y}_{\mathrm{pred}}\right)^2 \tag{4}$$

Here, the vectors $\vec{y}_{\mathrm{obs}}$ and $\vec{y}_{\mathrm{pred}}$ represent one observed time series and the corresponding prediction. If several time series contribute to the optimization, their differences (residuals) can be summed up:

$$L = \sum_{n=0}^{N} \left(\vec{y}_{\mathrm{obs,n}} - \vec{y}_{\mathrm{pred,n}}\right)^2 \tag{5}$$

To keep the notation simple, we will in the following use $Y_{\text{obs}}$ and $Y_{\text{pred}}$ to refer to the set of $N$ time series vectors. While each individual pair of $\vec{y}_{\text{pred,n}}$ and $\vec{y}_{\text{obs,n}}$ vectors must have the same length, note that different pairs might be of different length. $Y$ should thus not be interpreted as a matrix notation. In later chapters, we will see how the Python implementation handles the sets of observations (Section 3.2.4).

Coming back to the likelihood functions introduced in Section 2.1, the residual-based loss functions are a special case of a broader estimation concept, the maximum likelihood estimation:

$$\vec{\theta}_{\text{MLE}} = \underset{\vec{\theta}}{\text{argmax}} \ \ \mathcal{L}(\vec{\theta} \mid Y_{\text{obs}}) \tag{6}$$

Here, a probability density function is used to quantify how well observation and prediction, the latter represented by the model parameters, match. In case of a Normal-distributed likelihood with constant noise, the result of MLE is the same as a weighted least-squares loss [20]. In comparison to residual-based approaches, the choice of the PDF in a likelihood approach leads to more flexibility, for example covering heteroscedasticity or measurement noise that cannot be described by a Normal distribution.

As introduced in Section 2.1, an important extension of the likelihood approach is Bayes' theorem (Eq 1). Applying this concept, we can perform Bayesian inference of model parameters:

$$p(\vec{\theta} \mid Y_{\text{obs}}) = \frac{p(\vec{\theta}) \cdot \mathcal{L}(\vec{\theta} \mid Y_{\text{obs}})}{\int p(\vec{\theta}) \cdot \mathcal{L}(\vec{\theta} \mid Y_{\text{obs}}) \ d\vec{\theta}} \tag{7}$$

$$\vec{\theta}_{\text{MAP}} = \underset{\vec{\theta}}{\text{argmax}} \ \ p(\vec{\theta} \mid Y_{\text{obs}}) \tag{8}$$

Similar to MLE, a point estimate of the parameter vector with highest probability can be obtained by optimization (Eq 8), resulting in the maximum a posteriori (MAP) estimate. While the MLE is focused on the data-based likelihood, MAP estimates incorporate prior knowledge $p(\vec{\theta})$ into the parameter estimation.

To obtain the full posterior distribution $p(\vec{\theta} \mid Y_{\text{obs}})$, which is describing the probability distribution of parameters given the observed data, one has to solve Eq 7. The integral, however, is often intractable or impossible to solve analytically. Therefore, a class of algorithms called Markov chain Monte Carlo (MCMC) algorithms is often applied to find numerical approximations for the posterior distribution (for more detail, see Section 3.2.6).

The possibility to not only obtain point estimates but a whole distribution describing the parameter vector, is leading to an important concept: uncertainty quantification.

## 2.3 Uncertainty quantification of model parameters

When aiming for predictive models, it is important to not only estimate one parameter vector, but to quantify how certain the estimation is. In the frequentist paradigm, uncertainty is quantified with confidence intervals. When applied correctly, they provide a useful measure, for example in hypothesis testing where the size of a certain effect in a study is to be determined. However, interpretation of the confidence interval can be challenging and it is frequently misinterpreted as the interval that has a 95% chance to contain the true effect size or true mean [11]. However, to obtain intervals with such a simple interpretation, further assumptions on model parameters are required [12].

In Bayesian inference, prior distribution provide these necessary assumptions and the posterior can be used for uncertainty quantification. As a consequence, Bayesian credible intervals can indeed be interpreted as the range in which an unobserved parameter lies with a certain probability [13]. The choice of probability level or interval bounds is arbitrary. Commonly chosen probability levels are 99, 95, 94 or 90%. Consequently, there are many equally valid flavors of credible intervals. The most important ones are listed below:

- **Highest posterior density** intervals (HDI) are chosen such that the width of the interval is minimized

- **Equal-tailed** intervals (ETI) are chosen such that the probability mass of the posterior below and above the interval are equal

- **Half-open** credible intervals specify the probability that the parameter lies on one side of a threshold

In the scope of this paper, we will solely focus on the Bayesian quantification of parameter uncertainty. Note that uncertainty of parameters should not be confused with the measurement uncertainty mentioned in the context of calibration in Section 1.2, which will be further explained in the following section.

## 2.4 Calibration models

Coming back to the BIPM definition of calibration (Section 1.1), we can now associate aspects of that definition with the statistical modeling terminology. In Fig 2, the blue axis "independent" variable corresponds to the "quantity values" from the BIPM definition. At every value

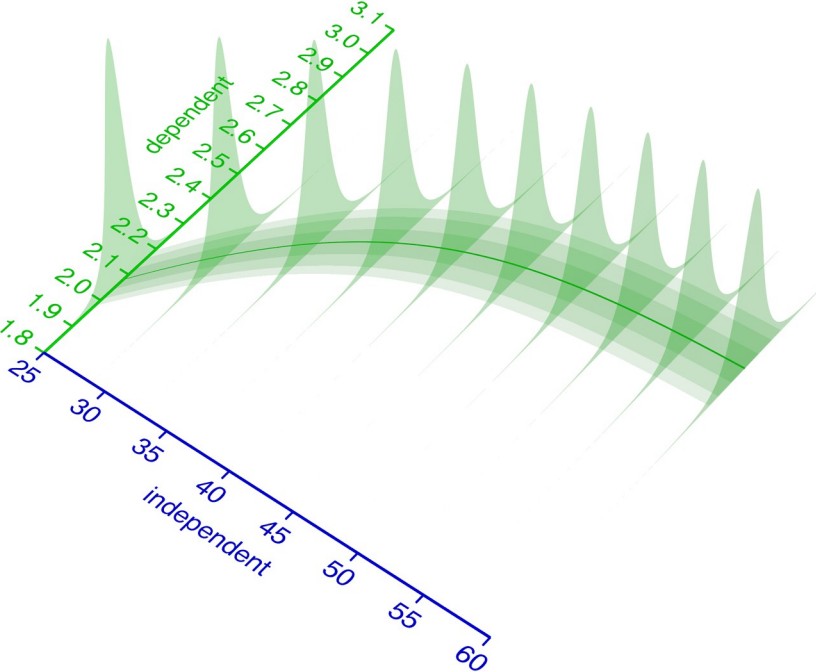

**Fig 2. Relationship of independent and dependent variable.** The distribution of measurement responses (dependent variable) can be modeled as a function of the independent variable. This measurement response probability distribution (here: Student-$t$) is parametrized by its parameters the mean $\mu$ (solid green line) and spread parameters $\sigma$ and $\nu$. Some or all of the distributions parameters are modeled as a function of the independent variable.

of the independent variable, the calibration model (green) describes the probability distribution (green slices) of measurement responses. This corresponds to the "indications with associated measurement uncertainties" from the BIPM definition.

Neither the formal definition nor the conceptual framework presented in this study impose constraints on the kind of probability distribution that describes the measurement responses. Apart from the Normal distribution, a practitioner may choose a Student-$t$ distribution if outliers are a concern. The Student-$t$ distribution has a $\nu$ parameter that influences how much probability is attributed to the tails of the distribution (S1 Fig), or in other words how likely it is to observe extreme values. Other distributions such as Laplace or Huber distributions were also shown to be beneficial for outlier-corrupted data [21]. Depending on the measurement system at hand, a Lognormal, Gamma, Weibull, Uniform or other continuous distributions may be appropriate. Also discrete distributions such as the Poisson, Binomial or Categorical may be chosen to adequately represent the observation process. A corresponding example with a Poisson distribution for the dependent variable is included in the documentation [22].

For some distributions, including Normal and Student-$t$, the parameters may be categorized as location parameters affecting the median or spread parameters affecting the variance, while for many other distributions the commonly used parameterization is not as independent. The parameters of the probability distribution that models the measurement responses must be described as functions of the independent variable. In the example from Fig 2 relationship, a Student-$t$ distribution with parameters $\{\mu, \text{scale}, \nu\}$ is used. Its parameter $\mu$ is modeled with a logistic function, the scale parameter as a 1st order polynomial of $\mu$ and $\nu$ is kept constant. It should be emphasized that the choice of probability distribution and functions to model its parameters is completely up to the domain expert.

When coming up with the structure of a calibration model, domain knowledge about the measurement system should be considered, particularly for the choice of probability distribution. An exploratory scatter plot can help to select an adequate function for the location parameter of the distribution ($\mu$ in case of a Normal or Student-$t$). A popular choice for measurement systems that exhibit saturation kinetics is the (asymmetric) logistic function. Many other measurement systems can be operated in a "linear range", hence a 1st order polynomial is an equally popular model for the location parameter of a distribution. To describe the spread parameters ($\sigma$, scale, $\nu$, . . .), a 0th (constant) or 1st order (linear) polynomial function of the location parameter is often a reasonable choice.

After specifying the functions in the calibration model, the practitioner must fit the model (Section 2.2) and decide to stick with the result, or modify the functions in the model. This iteration between model specification and inspection is a central aspect of modeling. To avoid overfitting or lack of interpretability, we recommend to find the simplest model that is in line with domain knowledge about the measurement system, while minimizing the lack-of-fit.

The term lack-of-fit is used to describe systematic deviation between the model fit and data. It refers not only to the trend of location and spread parameters but also to the kind of probability distribution. A residual plot is often instrumental to diagnose lack-of-fit and discriminate it from purely random noise in the observations. In Fig 3, different calibration models (top), residuals (middle) and the spread of data points along the percentiles of the probability distribution (bottom) illustrate how to diagnose a lack-of-fit. The blue data points in Fig 3C are Lognormal-distributed, but the calibration model assumes a Student-$t$ distribution. In such cases, a large number of observations may be needed to spot that the chosen noise model does match the data. As an alternative to a percentile-based visualization, practitioners may therefore opt to perform a Kolmogorov-Smirnov test, or visualize based on (empirical) cumulative density functions (ECDF). Ideally such a visualization should be prepared with ECDF confidence bands [23] and we invite the interested reader to contribute an implementation of this method

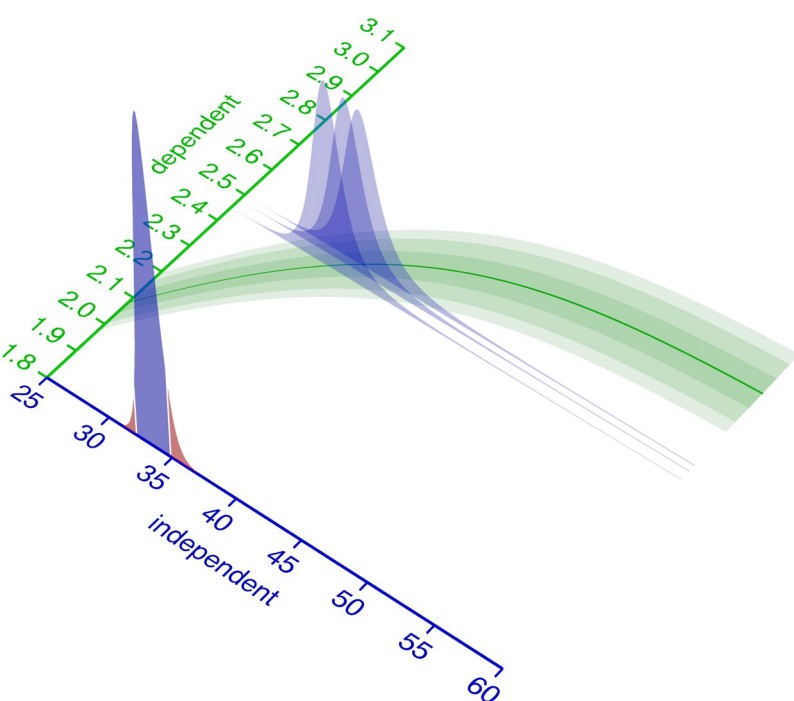

**Fig 3. Diagnostic plots of model fits.** The raw data (blue dots) and corresponding fit is visualized in the top row alongside 95, 90, and 68% likelihood bands of the model. Linear and logistic models were fitted to synthetic data to show three kinds of lack-of-fit error (columns 1–3) in comparison to a perfect fit (column 4). The underlying structure of the data and model is as follows: **A**: Homoscedastic linear model, fitted to homoscedastic nonlinear data. **B**: Homoscedastic linear model, fitted to heteroscedastic linear data. **C**: Homoscedastic linear model, fitted to homoscedastic linear data that is Lognormal-distributed. **D**: Heteroscedastic logistic model, fitted to heteroscedastic logistic data. The residual plots in the middle row show the distance between the data and the modeled location parameter (green line). The bottom row shows how many data points fall into the percentiles of the predicted probability distribution. Whereas the lack-of-fit cases exhibit systematic under- and over-occupancy of percentiles, only in the perfect fit case all percentiles are approximately equally occupied.

to `calibr8`. A well-chosen model (Fig 3D) is characterized by the random spread of residuals without systematic deviation and the equivalence of the modeled and observed distribution. When enough calibration data points are available, the modeled and observed distributions can be compared via the occupancy of percentiles.

Whereas the BIPM definition uses the word uncertainty in multiple contexts, we prefer to always use the term to describe uncertainty in a parameter, but never to refer to measurement noise. In other words, the parameter uncertainty can often be reduced by acquiring more data, whereas measurement noise is inherent and constant. In the context of calibration models, the methods for uncertainty quantification (Section 2.3) may be applied to the calibration model parameters, the independent variable, or both. Uncertainty quantification of calibration model parameters can be useful when investigating the structure of the calibration model itself, or when optimization does not yield a reliable fit. Because the independent variable is in most cases the parameter of interest in the application of a calibration model, the quantification of uncertainty about the independent variable is typically the goal. To keep the examples easy and understandable, we fix calibration model parameters at their maximum likelihood estimate. The `calibr8` documentation includes an example where the calibration model parameters are estimated jointly together with the process model [22].

In Fig 4 uncertainty, the green likelihood bands on the ground of the 3D plot represent a calibration model with fixed parameters. To quantify the independent variable with

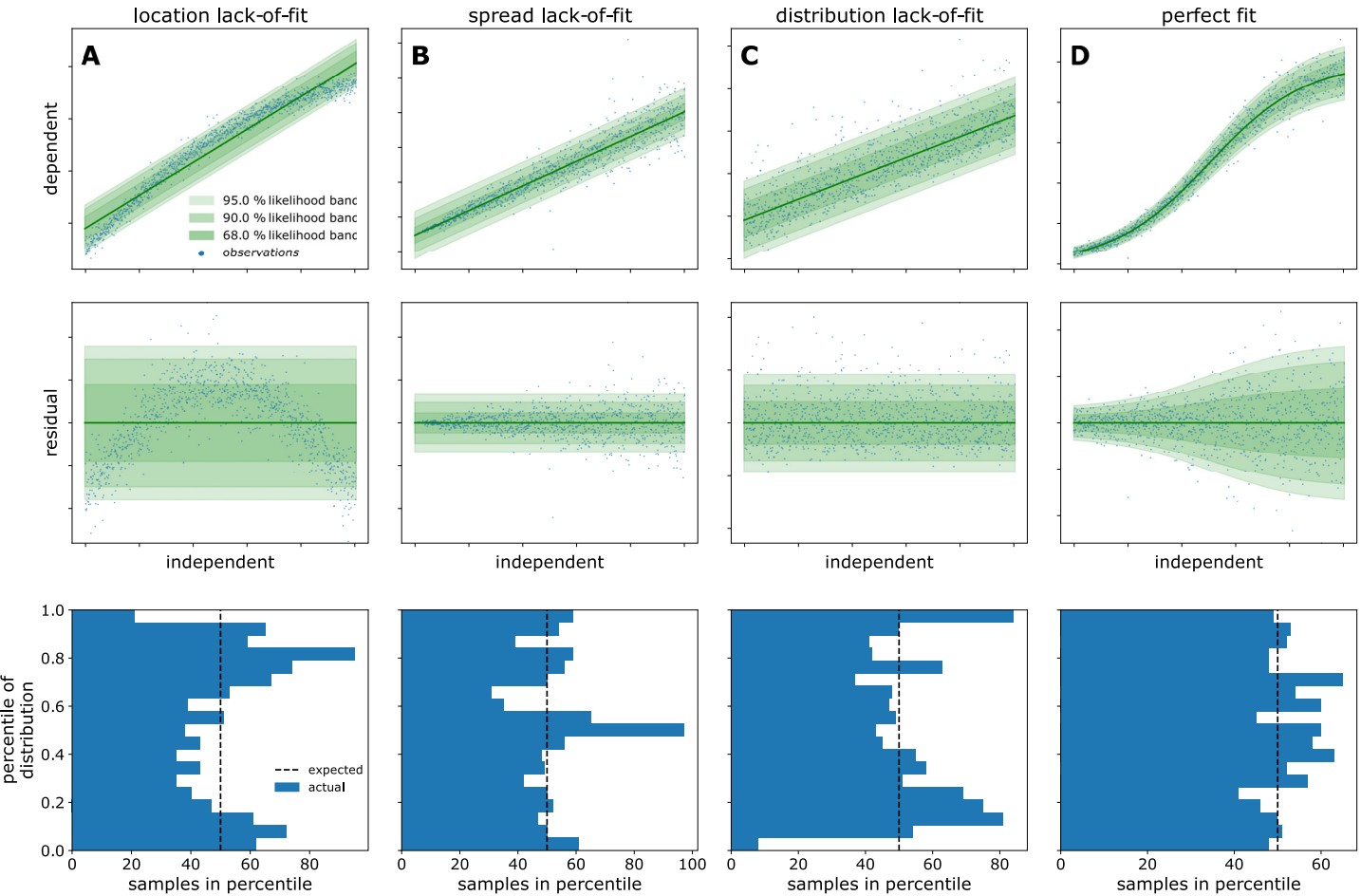

**Fig 4. Uncertainty about the independent variable.** An intuition for inferring the independent variable from an observed dependent variable is to cut (condition) the green probability distribution model at the observed value (blue slices) and normalize its area to 1. The resulting (blue) slice is a potentially asymmetric probability distribution that describes the likelihood of the observation, given the independent variable. Its maximum (the maximum likelihood estimate) is the value of the independent variable that best describes the observation. For multiple observations, the probability density function for the independent variable corresponds to the product of the PDFs of the observations. The red shoulders mark the regions outside of the 90% equal-tailed interval.

associated Bayesian uncertainty, it must be considered as a random variable. Accordingly, $p(\text{rv}_{\text{independent}} \mid \text{d})$ from either a likelihoodist (Eq 2) or Bayesian (Eq 1) perspective is the desired outcome of the uncertainty quantification.

Given a single observed dependent variable, the likelihoodist $p(\text{rv}_{\text{independent}} \mid d)$ (Eq 2) corresponds to the normalized cross section of the likelihood bands at the observed dependent variable (Fig 4, blue slices). With multiple observations, $p(\text{rv}_{\text{independent}} \mid d)$ becomes the product (superposition) of the elementwise likelihoods (Fig 4, blue slice at the axis). For a Bayesian interpretation of $p(\text{rv}_{\text{independent}} \mid d)$ (Eq 1), the blue likelihood slice is superimposed with an additional prior distribution. More practical details on uncertainty quantification of the independent variable in a calibration model are given in Section 4.

## 2.5 Process models

Most research questions are not answered by inferring a single variable from some observations. Instead, typical questions target the comparison between multiple conditions, the value of a latent (unobservable) parameter, or the inter- and extrapolation of a temporally evolving

system. For example, one might extract a latent parameter that constitutes a key performance indicator, or make decisions based on predictions (extrapolation) of new scenarios. Data analysis for all of these and many more scenarios is carried out with models that are tailored to the system or process under investigation. Such models are typically derived from theoretical (textbook) understanding of the process under investigation and in terms of SI units, but are not concerned with the means of making observations. Henceforth, we use the term process model ($\phi_{pm}$) to describe such models and discriminate them from calibration models ($\phi_{cm}$) that are explicitly concerned with the observation procedure.

Whereas calibration models are merely univariate input/output relationships of a measurement system, process models may involve many parameters, hierarchy, multivariate predictions or more complicated functions such as ordinary or partial differential equations. For example, they may predict a temporal evolution of a system with differential equations, sharing some parameters between different conditions, while keeping others local. In life sciences, time series play a central role, hence our application example is also concerned with a temporally evolving system.

Nevertheless, calibration models $\phi_{cm}$ and process models $\phi_{pm}$ are models, and the methods for estimation of their parameters (Section 2.2) as well as uncertainty quantification (Section 2.3) apply to both. As described in Section 2.3, the likelihood $\mathcal{L}$ is the ultimate allrounder tool in parameter estimation. The key behind our proposed discrimination between calibration and process models is the observation that a calibration model can serve as a modular likelihood function for a process model (Eq 9).

$$
\begin{aligned}
\hat{Y}_{pm} &= \phi_{pm}(\vec{\theta}_{pm}) \\
\hat{Y}_{cm} &= \phi_{cm}(\hat{Y}_{pm}, \vec{\theta}_{cm}) \\
\mathcal{L}(\vec{\theta}_{pm}, \vec{\theta}_{cm} \mid Y_{obs}) &= \mathcal{L}(\hat{Y}_{cm} \mid Y_{obs}) \\
&= p(Y_{obs} \mid \hat{Y}_{cm})
\end{aligned}
\tag{9}
$$

Conceptually separating between calibration models and process models has many advantages for the data analysis workflow in general. For example, the model components are logically separated, but the parameters can still be jointly estimated. After going into more detail about the implementation of calibration models and process models in Section 3, we will demonstrate their application and combination in Section 4.

## 3 Material and methods

### 3.1 Experimental workflows

**3.1.1 Automated cultivation platform.**   All experiments were conducted on a robotic platform with an integrated small-scale cultivation system. In our setup, a BioLector Pro microbioreactor system (m2p-labs GmbH, Baesweiler, Germany), is integrated into a Tecan Freedom EVO liquid handling robot (Tecan, Männedorf, Switzerland). The BioLector Pro is a device to quasi-continuously observe biomass, pH and dissolved oxygen (DO) during cultivation of microorganisms in specialized microtiter plates (MTPs). These rectangular plates comprise multiple reaction cavities called "wells", usually with volumes in microliter or milliliter scale. The BioLector allows to control temperature and humidity while shaking the plates at adjustable frequencies between 800 and 1500 rpm.

The liquid handler, which allows to take samples for *at-line* measurements during cultivation, is surrounded by a laminar flow hood to ensure sterile conditions for liquid transfer operations. Next to the BioLector Pro, various other devices are available on the platform,

including an Infinite M Nano+ microplate photometer (Tecan, Männedorf, Switzerland), a cooling carrier and a Hettich Rotanta 460 robotic centrifuge (Andreas Hettich GmbH & Co. KG, Tuttlingen, Germany). The overall setup is similar to the one described by Unthan et al. 2015 [8]. The automated platform enables to perform growth experiments with different microorganisms, to autonomously take samples of the running process and to perform bioanalytical measurements, e.g. quantification of glucose. It is thus a device for miniaturised, automated bioprocess cultivation experiments.

In this work, we used specialized 48-well deepwell plates called FlowerPlates (MTP-48-B, m2p-labs GmbH, Baesweiler, Germany) that are commonly used for cultivations in the BioLector device. The plates are pre-sterilized and disposable and were covered with a gas-permeable sealing film with a pierced silicone layer for automation (m2p-labs GmbH, Baesweiler, Germany). The biomass was quasi-continuously detected via scattered light [24] at gain 3 with 4 minutes cycle time to obtain backscatter measurements. DO and pH were not measured since they are not relevant for the application examples. Both cultivation and biomass calibration experiments were conducted in the BioLector Pro at 30˚C, 3 mm shaking diameter, 1400 rpm shaking frequency, 21% head-space oxygen concentration and $\geq$ 85% relative humidity.

**3.1.2 Strain, media preparation and cell banking and cultivation.**   The wild-type strain *Corynebacterium glutamicum* ATCC 13032 [25] was used in this study. If not stated otherwise, all chemicals were purchased from Sigma–Aldrich (Steinheim, Germany), Roche (Grenzach-Wyhlen, Germany) or Carl Roth (Karlsruhe, Germany) in analytical quality.

Cultivations were performed with CGXII defined medium with the following final amounts per liter of distilled water: 20 g D-glucose, 20 g $(NH_4)_2SO_4$, 1 g $K_2HPO_4$, 1 g $KH_2PO_4$, 5 g urea, 13.25 mg $CaCl_2 \cdot 2\,H_2O$, 0.25 g $MgSO_4 \cdot 7\,H_2O$, 10 mg $FeSO_4 \cdot 7\,H_2O$, 10 mg $MnSO_4 \cdot H_2O$, 0.02 mg $NiCl_2 \cdot 6\,H_2O$, 0.313 mg $CuSO_4 \cdot 5\,H_2O$, 1 mg $ZnSO_4 \cdot 7\,H_2O$, 0.2 mg biotin, 30 mg protocatechuic acid. $42\frac{g}{L}$ MOPS were used as buffering agent and the pH was adjusted to 7.0 using 4 M NaOH.

A working cell bank (WCB) was prepared from a shake flask culture containing 50 mL of the described CGXII medium and 10% (v/v) brain heart infusion (BHI) medium ($37\frac{g}{L}$). It was inoculated with 100 µl cryo culture from a master cell bank stored at -80˚C. The culture was incubated for approximately 16 hours in an unbaffled shake flask with 500 ml nominal volume at 250 rpm, 25 mm shaking diameter and 30˚C. The culture broth was then centrifuged at $4000 \times g$ for 10 minutes at 4˚C and washed once in 0.9% (w/v) sterile NaCl solution. After centrifugation, the pellets were resuspended in a suitable volume of NaCl solution to yield a suspension with an optical density at 600 nm ($OD_{600}$) of 60. The suspension was then mixed with an equal volume of 50% (w/v) sterile glycerol, resulting in cryo cultures of $OD_{600}\approx30$. Aliquots of 1 mL were quickly transferred to low-temperature freezer vials, frozen in liquid nitrogen and stored at -80˚C.

**3.1.3 Algorithmic planning of dilution series.**   All calibration experiments require a set of standards (reference samples) with known concentrations, spanning across sometimes multiple orders of magnitude. Traditionally such standards are prepared by manually pipetting a serial dilution with a 2x dilution factor in each step. This can result in a series of standards whose concentrations are evenly spaced on a logarithmic scale. While easily planned, a serial dilution generally introduces inaccuracies that accumulate with an increasing number of dilution steps. It is therefore desirable to plan a dilution series of reference standards such that the number of serial dilution steps is minimized.

To reduce the planning effort and allow for a swift and accurate preparation of the standards, we devised an algorithm that plans liquid handling instructions for preparation of standards. Our `DilutionPlan` algorithm considers constraints of a ($R \times C$) grid geometry, well

volume, minimum and maximum transfer volumes to generate pipetting instructions for human or robotic liquid handlers.

First, the algorithms reshapes a length $R \cdot C$ vector of sorted target concentrations into the user specified $(R \times C)$ grid typically corresponding to a microtiter plate. Next, it iteratively plans the transfer and filling volumes of grid columns which are subject to the volume constraints. This column-wise procedure improves the compatibility with multi-channel manual pipettes, or robotic liquid handlers. Diluting from a stock solution is prioritized over the (serial) dilution from already diluted columns. The result of the algorithm are (machine-readable) pipetting instructions to create $R \cdot C$ single replicates with concentrations very close to the targets. We open-sourced the implementation as part of the `robotools` library [26].

As the accuracy of the calibration model parameter estimate increases with the number of calibration points, we performed all calibrations with the maximum number of observations that the respective measurement system can make in parallel. The calibration with 96 glucose and 48 biomass concentrations is covered in the following chapters.

**3.1.4 Glucose assay calibration.** For the quantification of D-glucose, the commercial enzymatic assay kit "Glucose Hexokinase FS" (DiaSys Diagnostic Systems, Holzheim, Germany) was used. For the master mix, four parts buffer and one part enzyme solution were mixed manually. The master mix was freshly prepared for each experiment and incubated at room temperature for at least 30 minutes prior to the application for temperature stabilization. All other pipetting operations were performed with the robotic liquid handler. For the assay, 280 μL master mix were added to 20 μL analyte in transparent 96-well flat bottom polystyrol plates (Greiner Bio-One GmbH, Frickenhausen, Germany) and incubated for 6 minutes, followed by absorbance measurement at 365 nm. To treat standards and cultivation samples equally, both were diluted by a factor of 10 (100 μL sample/standard + 900 μL diluent) as part of the assay procedure.

As standards for calibration, 96 solutions with concentrations between 0.075 and $50\frac{g}{L}$ were prepared from fresh CGXII cultivation medium (Section 3.1.2) with a $50\frac{g}{L}$ concentration of D-glucose. The `DilutionPlan` algorithm (Section 3.1.3) was used to plan the serial dilution procedure with glucose-free CGXII media as the diluent, resulting in 96 unique concentrations, evenly distributed on a logarithmic scale. Absorbance results from the photometer were parsed with a custom Python package (not published) and paired with the concentrations from the serial dilution series to form the calibration dataset used in Section 4.1.2. 83 of the 96 concentration/absorbance pairs lie below $20\frac{g}{L}$ and were used to fit a linear model in Section 4.1.1.

**3.1.5 Biomass calibration.** Calibration data for the biomass/backscatter calibration model (Section 4.1.2) was acquired by measuring 48 different biomass concentrations at cultivation conditions (Section 3.1.2) in the BioLector Pro. 100 mL *C. glutamicum* WT culture was grown overnight on $20\frac{g}{L}$ glucose CGXII medium (Section 3.1.2) in two unbaffled 500 mL shake flasks with 50 mL culture volume each (N = 250 rpm, r = 25 mm). The cultures were harvested in the stationary phase, pooled, centrifuged and resuspended in 25 mL 0.9% (w/v) NaCl solution. The highly concentrated biomass suspension was transferred into a magnetically stirred 100 mL trough on the liquid handler for automated serial dilution with logarithmically evenly spaced dilution factors from 1× to 1000×. The serial dilution was prepared by the robotic liquid handler in a 6 × 8 (48-well square) deep well plate (Agilent Part number 201306–100) according to the `DilutionPlan` (Section 3.1.3). 6x 800 μL of biomass stock solution were transferred to previously dried and weighed 2 mL tubes, immediately after all transfers of stock solution to the 48 well plate had occurred. The 2 mL tubes were frozen at -80˚C, lyophilized over night, dried again at room temperature in a desiccator over night and finally weighted again to determine the biomass concentration in the stock solution.

After a column in the 48 well plate was diluted with 0.9% (w/v) NaCl solution, the column was mixed twice by aspirating 950 μL at the bottom of the wells and dispensing above the liquid surface. The transfers for serial dilutions (columns 1 and 2) and to the 48 well FlowerPlate were executed right after mixing to minimize the effects of biomass sedimentation as much as possible. The FlowerPlate was sealed with a gas-permeable sealing foil (product number F-GP-10, m2p-labs GmbH, Baesweiler, Germany) and placed in the BioLector Pro device. The 1 h BioLector process for the acquisition of calibration data was programmed with shaking frequency profile of 1400, 1200, 1500, 1000 rpm while maintaining 30˚C chamber temperature and measuring backscatter with gain 3 in cycles of 3 minutes.

The result file was parsed with the `bletl` Python package [27] to extract backscatter measurements made at 1400 rpm shaking frequency. A log(independent) asymmetric logistic calibration model was fitted as described in Section 4.1.2. The linear calibration model for comparison purposes (Section 4.2.3) was implemented with its intercept fixed to the background signal predicted by the asymmetric logistic model ($\mu_{\mathrm{BS}}\left(0\ \frac{g}{L}\right)$). It was fitted to a subset of calibration points approximately linearly spaced at 30 different biomass concentrations from 0.01 to $15\frac{g}{L}$.

**3.1.6 Microbial growth experiment.**   Cultivations with *C. glutamicum* were performed in the automated cultivation platform (Section 3.1.1) under the described conditions. CGXII medium with $20\frac{g}{L}$ glucose and without BHI was used as cultivation medium. To start the growth experiments, the liquid handler was used to transfer 20 μL of a WCB aliquot into the first column of FlowerPlate wells, which were pre-filled with 780 μL medium. These wells were run as a preculture. When precultures reached a backscatter readout of 15, which corresponds to a cell dry weight of approximately $10\frac{g}{L}$, the inoculation of the main culture wells was triggered. 780 μL medium were distributed into each main culture well (columns 2–8) and allowed to warm up for approximately 15 minutes. Preculture wells A01 and B01 were pooled and 20 μL culture broth was transferred to each main culture well, resulting in 800 μL final volume. The theoretical biomass concentration at the start of the main cultures is $0.25\frac{g}{L}$ accordingly. This strategy was used to avoid a lag-phase with non-exponential growth.

Backscatter measurements of biomass concentration were acquired continuously, while main culture wells were harvested at predetermined time points to measure glucose concentrations in the supernatant. The time points were chosen between 0 and 15 hours after the inoculation of main cultures to cover all growth phases. For harvesting, the culture broth was transferred to a 1 mL deep-well plate by the liquid handler. The plate was centrifuged at $3190 \times g$ at 4˚C for 5 minutes and the supernatant was stored on a 1 mL deep well plate chilled to 4˚C. The glucose assay was performed after all samples were taken.

## 3.2 Computational methods

All analyses presented in this study were performed with recent versions of Python 3.7, PyMC ==3.11.2 [28], ArviZ >=0.9 [29], PyGMO >=2.13 [30], matplotlib >=3.1 [31], NumPy >=1.16 [32], pandas >=0.24 [33, 34], SciPy >=1.3 [35] and related packages from the Python ecosystem. For a full list of dependencies and exact versions see the accompanying GitHub repository [36].

The two packages presented in this study, `calibr8` and `murefi`, may be installed via semantically versioned releases on PyPI. Source code, documentation and detailed release notes are available through their respective GitHub projects [37, 38].

**3.2.1 Asymmetric logistic function.**   The asymmetric, five-parameter logistic function (also known as Richard's curve) was previously shown to be a good model for many applications [39], but it is often defined in a parameterization (Eq 10) that is non-intuitive. Some parametrizations even introduce a sixth parameter to make the specification more intuitive,

but this comes at the cost of structural non-identifiability [40, 41]. Furthermore, in the most commonly found parametrization (Eq 10), one parameter is constrained to be strictly positive. We also found that structural non-identifiability between the parameters makes it difficult to define an initial guess and bounds to reliably optimize a model based on this parametrization.

$$f(x) \quad = L_L + \frac{L_U - L_L}{\left(1 + e^{-B(m-x)}\right)^{1/v}}$$

$$L_L, L_U, B, m \quad \in \mathbb{R}$$

$$v \quad \in \mathbb{R}_{>0}$$

(10)

To make calibration model fitting more user-friendly, we reparameterized the commonly used form such that all five parameters are intuitively interpretable and structurally independent (Fig 5). With our reparameterization (Eq 11), the 5-parameter asymmetric logistic function is parameterized by lower limit $L_L \in \mathbb{R}$, upper limit $L_U \in \mathbb{R}$, inflection point x-coordinate $I_x \in \mathbb{R}$, slope at inflection point $S \in \mathbb{R}$ and an asymmetry parameter $c \in \mathbb{R}$. At $c = 0$, the y-coordinate of the inflection point lies centered between $L_L$ and $L_U$. $I_y$ moves closer to $L_U$ when $c > 0$ and accordingly closer to $L_L$ when $c < 0$ (Fig 5, black and gray curves). An interactive version of Fig 5 can be found in a Jupyter Notebook in the `calibr8` GitHub repository [37].

For readability and computational efficiency, we used `SymPy` [42] to apply common subexpression elimination to Eq 11 and our implementation respectively (S1 File). The step wise derivation from Eqs 10 to 11 is shown in S1 File and in a Jupyter Notebook in the `calibr8` GitHub repository [37].

$$f(x) \quad = L_L + \frac{L_U - L_L}{\left(e^{s_2 \cdot \left(s_3 \cdot (I_x - x) + \frac{c}{s_2}\right)} + 1\right)^{s_1}}$$

$$s_0 \quad = e^c + 1$$

$$s_1 \quad = e^{-c}$$

$$s_2 \quad = s_0^{(s_0 \cdot s_1)}$$

$$s_3 \quad = \frac{S}{L_U - L_L}$$

$$L_L, L_U, I_x, S, c \quad \in \mathbb{R}$$

(11)

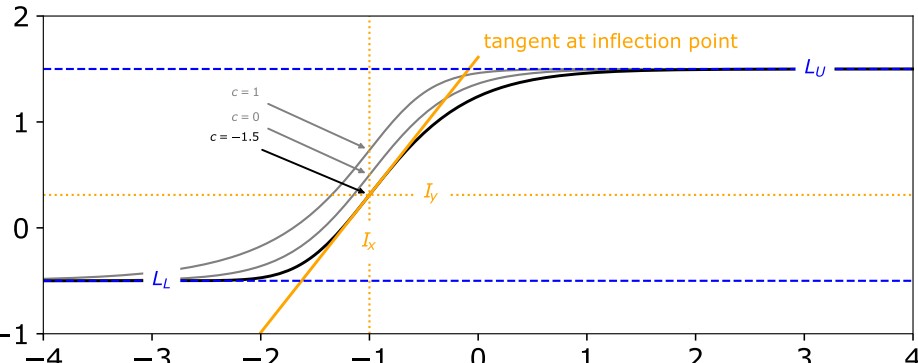

**Fig 5. Reparametrized asymmetric logistic function.** When parametrized as shown in Eq 11, each of the 5 parameters can be manipulated without influencing the others. Note that, for example, the symmetry parameter $c$ can be changed without affecting the x-coordinate of the inflection point ($I_x$), or the slope $S$ at the inflection point (gray vs. black).

**3.2.2 `calibr8` package for calibration models and modular likelihoods.** With `calibr8` we present a lightweight Python package that specializes on the definition and modular implementation of non-linear calibration models for calibration and modeling workflows.

The `calibr8` application programming interface (API) was designed such that all calibration models are implemented as classes that inherit from `calibr8.CalibrationModel`, which implements properties and methods that are common to all calibration models (Fig 6).

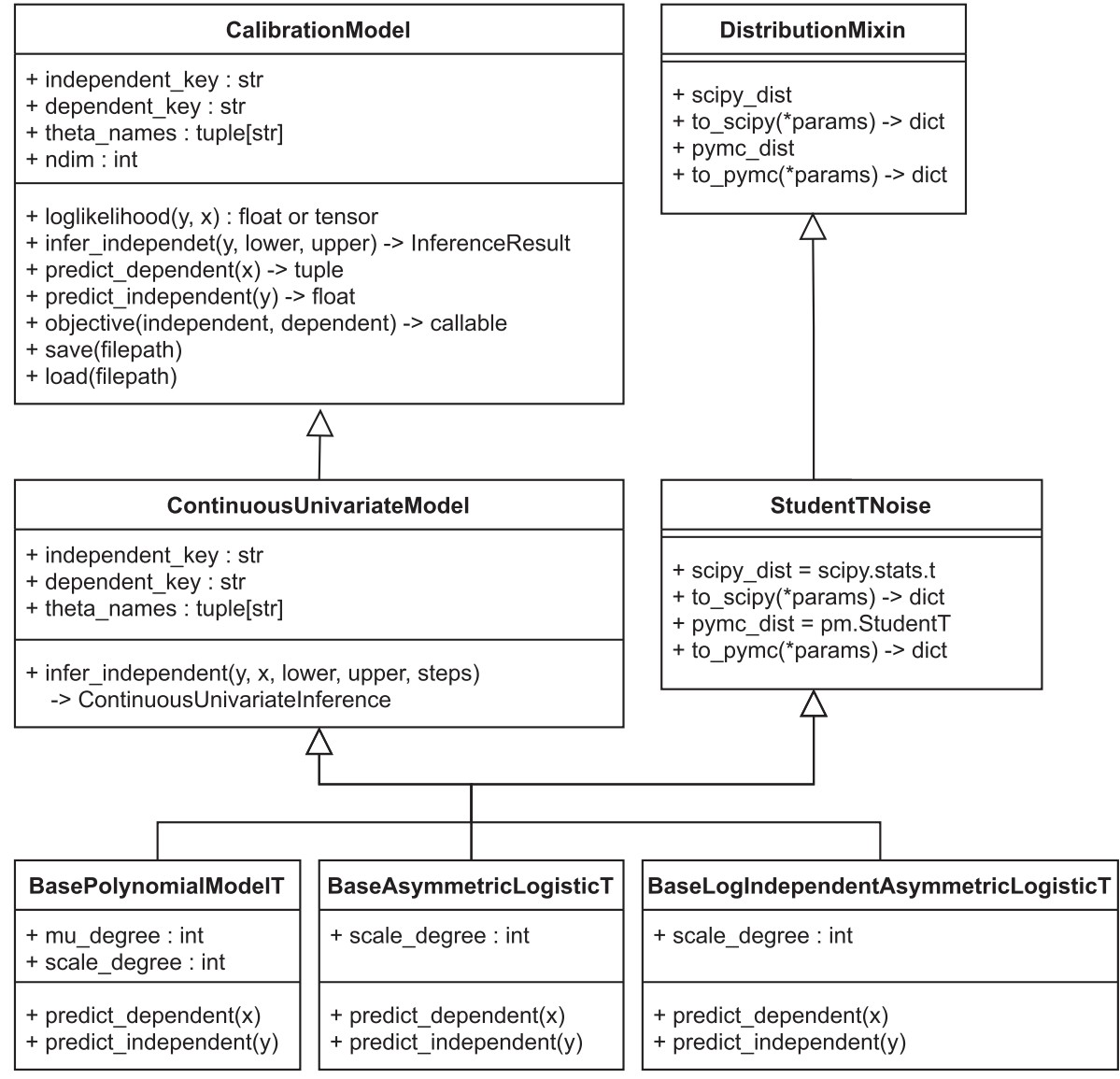

**Fig 6. `calibr8` class diagram.** All `calibr8` models inherit from the same `CalibrationModel` and `DistributionMixin` classes that define attributes, properties and method signatures that are common to all calibration models. Some methods, like `loglikelihood()` or `objective()` are implemented by `CalibrationModel` directly, whereas others are implemented by the inheriting classes. Specifically the `predict_*` methods depend on the choice of the domain expert. With a suite of `Base*T` classes, `calibr8` provides base classes for models based on Student-*t* distributed observations. A domain expert may start from any of these levels to implement a custom calibration model for a specific application.

Box 1. Code 1. Implementation of glucose/absorbance calibration model using convenience type.

```
1  class LinearGlucoseCalibrationModelV1(calibr8.BasePolynomialModelT):
2      def __init__(self, *, independent_key:str="S", dependent_key:str="A365"):
3          super().__init__(
4              independent_key=independent_key,
5              dependent_key=dependent_key,
6              mu_degree=1,
7              scale_degree=1
8          )
```

The common interface simplifies working with calibration models in a data analysis or modeling workflow. For a detailed explanation of classes and inheritance in `calibr8` we refer the interested reader to the documentation [22]. For example, the `CalibrationModel.objective` can be used to create objective functions to optimize the model parameters. The objective relies on the `loglikelihood` method to compute the sum of log-likelihoods from independent and dependent variables. It uses the `predict_dependent` method internally to obtain the parameters of the probability distribution describing the dependent variables, conditioned on the independent variable.

Through its inheritance-based design, the `calibr8.CalibrationModel` gives the domain expert full control over the choice of trend functions and probability distributions. Conveniently, `calibr8` already implements functions such as `polynomial`, `logistic` and `asymmetric_logistic`, as well as base classes for commonly found noise models. By leveraging these base models, the implementation of a user-defined calibration model reduces to just a few lines of code (Box 1 Code 1 and Box 2 Code 2). In the current version we implemented base classes for continuous univariate and multivariate calibration models and anticipated models with discrete independent variables.

The implementations depicted in Fig 6 are fully compatible with `aesara.Variable` inputs, resulting in `TensorVariable` outputs. Aesara is a graph computation framework that auto-differentiates computation graphs written in Python and compiles functions that evaluate with high performance [43]. This way, the `loglikelihood` function of a `CalibrationModel` can be auto-differentiated and compiled, thus facilitating efficient computation with optimization or gradient-based MCMC sampling algorithms

Box 2. Code 2. Implementation of CDW/backscatter calibration model using convenience type.

```
1  class BioLectorCDWBackscatterModelV1(calibr8.BaseLogIndependentAsymmetricLogisticT):
2      def __init__(self, *, independent_key:str='X', dependent_key:str='BS'):
3          super().__init__(
4              independent_key=independent_key,
5              dependent_key=dependent_key,
6              scale_degree=1
7          )
```

(Section 3.2.6). For more details about the implementation, please refer to the documentation and code of the `calibr8` package [37].

**Convenience features**. To facilitate modeling workflows, `calibr8` implements convenience functions for optimization (`fit_scipy`, `fit_pygmo`) and creation of diagnostic plots (`calibr8.plot_model`) as shown in Section 2.4 and Section 4.1.2. As explained in Section 2.4 the residual plot on the right of the resulting figure is instrumental to judge the quality of the model fit.

Standard properties of the model, estimated parameters and calibration data can be saved to a JSON file via the `CalibrationModel.save` method. The saved file includes additional information about the type of calibration model and the `calibr8` version number (e.g. S3 File) to support good versioning and data provenance practices. When the `CalibrationModel.load` method is called to instantiate a calibration model from a file, standard properties of the new instance are set and the model type and `calibr8` version number are checked for compatibility.

**3.2.3 Numerical inference.** To numerically infer the posterior distribution of the independent variable, given one or more observations, `infer_independent` implements a multi-step procedure. The three outputs of this procedure are a vector of posterior probability evaluations, densely resolved around the locations of high probability mass, and the bounds of the equal-tailed as well as the highest-density intervals (ETI, HDI) corresponding to a user-specified credible interval probability.

In the first step, the likelihood function is integrated in the user-specified interval [lower, upper] with `scipy.integrate.quad`. Second, we evaluate its cumulative density function (CDF) at 10 000 locations in [lower, upper] and determine locations closest to the $\mathrm{ETI}^{99.999\%}$. Next, we re-evaluate the CDF at 100 000 locations in the $\mathrm{ETI}^{99.999\%}$ to obtain it with sufficiently high resolution in the region of highest probability. Both ETI and HDI with the (very close to) user-specified `ci_prob` are obtained from the high-resolution CDF. Whereas the ETI is easily obtained by finding the CDF evaluations closest to the corresponding lower and upper probability levels, the HDI must be determined through optimization (Eq 12).

$$\mathrm{HDI} = [a, a+d] = \operatorname*{argmin}_{a,d} \begin{cases} \infty & \text{if } \mathrm{CDF}(a+d) - \mathrm{CDF}(a) < \texttt{ci\_prob} \\ d & \text{otherwise} \end{cases} \tag{12}$$

**3.2.4 `murefi` package for building multi-replicate ODE models.** Models of biochemical processes are traditionally set up to describe the temporal progression of an individual system, such as a reaction vessel. Experimental data, however, is commonly obtained from multiple reaction vessels in parallel, often run under different conditions to maximize information gain. This discrepancy between the prototypical model of the biological system and the heterogeneous experimental data to be fitted is typically resolved by replicating the model for all realizations of the biological system in the dataset. Along with the replication of the model, some model parameters may be kept global, while others can be local to a subset of the replicates, for example due to batch effects or different start conditions.

With a Python package we call `murefi` (multi-replicate fitting), we implemented data structures, classes and auxiliary functions that simplify the implementation of models for such heterogeneous time series datasets. It seamlessly integrates with `calibr8` to construct likelihood-based objective functions for optimization or Bayesian inference. To enable the application of efficient optimization or sampling algorithms, the use of automatic differentiation to obtain gradients of the likelihood w.r.t. input parameters is highly desirable. Various methods

for automatic differentiation of ODE models are available, but their efficiency is closely connected to the implementation and size of the model [44]. In `murefi` we implemented support for `sunode` [45], a recently implemented Python wrapper around the SUNDIALS suite of nonlinear and differential/algebraic equation solvers [46]. When used in the context of a PyMC model, a process model created with `calibr8` and `murefi` can therefore be auto-differentiated, solved, optimized and MCMC-sampled with particularly high computational efficiency.

**Structure of time series data and models**. To accommodate for the heterogeneous structure of time series experiments in biological applications, we implemented a data structure of three hierarchy levels. The `murefi.Timeseries` object represents the time and state vectors $\vec{t}, \vec{y}$ of a single state variable or observation time series. To allow association of state and observed variables via `calibr8` calibration models, the `Timeseries` is initialized with `independent_key` and `dependent_key`. Multiple `Timeseries` are bundled to a `murefi.Replicate`, which represents either the observations obtained from one reaction vessel, or the predictions made by a process model. Consequently, the `murefi.Dataset` aggregates replicates of multiple reaction vessels, or the model predictions made for them (Fig 7, center). To allow for a separation of data preprocessing and model fitting in both time and implementation, a `murefi.Dataset` can be saved as and loaded from a single HDF5 file [47, 48].

To describe a reaction system by a system of ODEs, a new class is implemented by subclassing the `murefi.BaseODEModel` convenience type. In the constructor of the class, the names and order of parameters and state variables are defined, whereas the differential equations are implemented in a `dydt` instance method. An example is shown in Box 3 Code 3 with the implementation of the Monod kinetics for microbial growth.

**Parameter mapping and objective function.** In addition to a `murefi` model instance, a `murefi.Dataset` and calibration models, a `murefi.ParameterMapping` must be defined to facilitate the creation of an objective function. This mapping specifies whether parameters are local or global and the rules with which they are shared between replicates. The

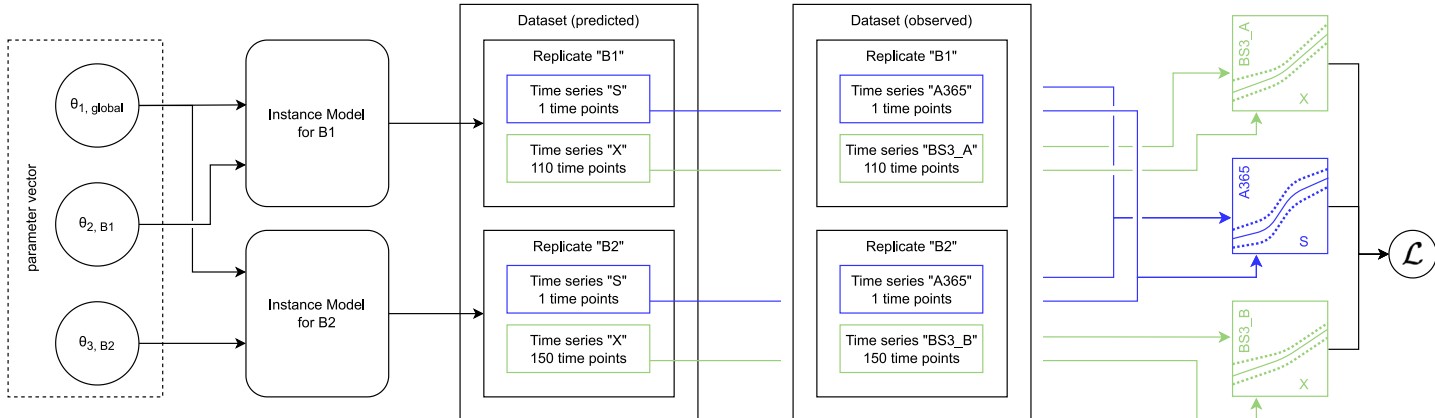

**Fig 7. Data structures and computation graph of `murefi` models.** Elements in a comprehensive parameter vector are mapped to replicate-wise model instances. In the depicted example, the model instances for both replicates "B1" and "B2" share $\theta_{1,global}$ as the first element in their parameter vectors. The second model parameter $\theta_2$ is local to the replicates, hence the full parameter vector (left) is comprised of three elements. Model predictions are made such that they resemble the structure of the observed data, having the same number of time points for each predicted time series. An objective function calculating the sum of log-likelihoods is created by associating predicted and observed time series via their respective calibration models. By associating the calibration models based on the dependent variable name, a calibration model may be re-used across multiple replicates, or kept local if, for example, the observations were obtained by different methods.

Box 3. Code 3. Implementation of Monod ODE model using `murefi.BaseODEModel` convenience type.

```
1  class MonodModel(murefi.BaseODEModel):
2      def __init__(self):
3          super().__init__(
4              theta_names=("S0", "X0", "mu_max", "K_S", "Y_XS"),
5              independent_keys=["S", "X"]
6          )
7
8      def dydt(self, y, t, theta):
9          S, X = y
10         mu_max, K_S, Y_XS = theta
11         dXdt = mu_max * S * X / (K_S + S)
12         dSdt = -1 / Y_XS * dXdt
13
14         return [
15             dSdt,
16             dXdt,
17         ]
```

`ParameterMapping` may be represented as a table, assigning each element of replicate-wise parameter vectors to constants or names of parameters in a comprehensive parameter vector. In Fig 7, the parameter mapping is depicted by arrows mapping elements of a 3-element comprehensive parameter vector to 2-element parameter vectors of the replicate-wise models. A table representation of the parameter mapping used to fit the Monod model in Section 4.2 is shown in S4 File.

Model predictions are made such that the time points of the predicted time series match those of the observed data (Fig 7, center). Based on the `(in)dependent_key`, the predicted and observed `Timeseries` can be associated with each other and passed to the corresponding `CalibrationModel.loglikelihood` method to calculate $\mathcal{L}(\vec{\theta} \mid Y_{\text{obs}})$. Note that this procedure conveniently allows for calibration models to be shared by multiple replicates, as well as making observations of one state variable with more than one analytical method.

An objective function performing the computation depicted in Fig 7 can be created with a single call to a convenience function. For compute-efficient optimization and specifically Bayesian parameter estimation, the elements in the parameter vector can also be Aesara tensors, resulting in the creation of a symbolic computation graph. The computation graph cannot only be statically compiled but also auto-differentiated, if all operations in the graph are also auto-differentiable. This is already the case for standard `calibr8` calibration models and is also available for `murefi` -based process models when the `sunode` [45] package is installed. A corresponding example of obtaining gradients from an ODE model with `calibr8`, `murefi` and `sunode` is included in the `murefi` documentation [49].

**3.2.5 Optimization.** In this work, optimization algorithms are involved at multiple steps of the workflow. Unless otherwise noted we used `scipy.optimize.minimize` with

default settings to obtain the MLEs of calibration and process models. Our current implementation to compute HDIs (Section 3.2.3) uses `scipy.optimize.fmin` with an underlying Nelder-Mead simplex algorithm.

Initial guesses, as well as parameter bounds for maximum-likelihood optimization, were motivated from prior assumptions or exploratory plots of the data. Based on the intuitive parametrization of the asymmetric logistic (Section 3.2.1) we specified initial guesses for calibration models such that the model prediction from the guessed parameter vector was at least in the same order of magnitude as the data. For MLE of process model parameters, the guessed parameters were motivated from prior assumptions, e.g. the amount of used substrate or values obtained from literature research. Likewise, we specified bounds to be realistic both biologically and based on exploratory scatter plots of the data.

**3.2.6 MCMC sampling.**    In contrast to optimization, MCMC sampling follows a very different paradigm. Whereas in MLE the likelihood function is iteratively evaluated to find its maximum, Bayesian inference aims to approximate the posterior probability distribution according to Eq 7.

Most sampling algorithms draw the posterior samples in the form of a Markov chain with a equilibrium distribution that matches the posterior probability distribution. While early MCMC algorithms, such as Random-walk Metropolis [50] are conceptually simple and easy to implement, they are computationally ineffective on problems with more than just a handful of dimensions [51, 52]. Instead of implementing inefficient algorithms by hand, practitioners can rely on state of the art libraries for Bayesian inference. These libraries apply automatic transformations, provide diagnostic tools and implement much more efficient sampling algorithms that often use gradients $\frac{d\mathcal{L}}{d\theta}$ for higher computational efficiency.

Probabilistic programming languages / libraries (PPL), such as PyMC [53], Pyro [54], Stan [55] or Tensorflow Probability [56] use automatic differentiation and typically implement at least one of the gradient-based sampling algorithms Hamiltonian Monte Carlo (HMC) or No-U-Turn Sampling (NUTS) [52]. While PPLs typically require a model to be implemented using their API, other libraries such as `emcee` [57] provide, for example, gradient-free ensemble algorithms that can be applied to black-box problems. PyMC, the most popular Python-based PPL, implements both gradient-based (HMC, NUTS) as well as gradient-free algorithms, such as Differential Evolution MCMC (DE-MCMC) [58], DE-MCMC-Z [51] or elliptical slice sampling [59] in Python, allowing easy integration with custom data processing and modeling code. In this study, PyMC was used to sample posterior distributions with either DE-MCMC-Z (`pymc.DEMetropolisZ`) or NUTS.

**MCMC sampling of the process model**. Whereas in DE-MCMC, proposals are informed from a random pair of other chains in a "population", the DE-MCMC-Z version selects a pair of states from its own history, the "Z"-dimension. Compared to DE-MCMC, DE-MCMC-Z yields good results with fewer chains that can run independently. The `pymc.DEMetropolisZ` sampler differs from the original DE-MCMC-Z in a tuning mechanism by which a `tune_drop_fraction` of by default 90% of the samples are discarded at the end of the tuning phase. This trick reliably cuts away unconverged "burn-in" history, leading to faster convergence.

`pymc.DEMetropolisZ` was applied to sample the process model in Section 4.2.3. MCMC chains were initialized at the MAP to accelerate convergence of the DE-MCMC-Z sampler in the tuning phase. 50 000 tuning iterations per chain were followed by 500 000 iterations to draw posterior samples for further analysis. The `DEMetropolisZ` settings remained fixed at ($\lambda = \frac{2.38}{\sqrt{2 \cdot d}}$ (default), $\epsilon = 0.0001$) for the entire procedure.

The $\hat{R}$ diagnostic from ArviZ [29] was used to check for convergence (all $\hat{R} \approx 1$, S1 Table).

**3.2.7 Visualization techniques.** Plots were prepared from Python with a combination of Matplotlib [31], ArviZ and PyMC. We used POV-Ray to produce Figs 2 and 4 and https://diagrams.net for technical drawings. Probability densities were visualized with the `pymc.gp.utils.plot_gp_dist` helper function that overlays many polygons corresponding to percentiles of a distribution, creating the colorful bands of plots seen in Section 4.2.4 and others. Posterior predictive samples were obtained by randomly drawing observations from the calibration model, based on independent values sampled from the posterior distribution. If not stated otherwise, the densities plotted for MCMC prediction results were obtained from at least 1000 posterior samples. The pair plots of 2-dimensional kernel density estimates of posterior marginals (e.g. S2 Fig) were prepared with ArviZ.

## 4 Results and discussion

### 4.1 Application: Implementing (non-)linear calibration models with `calibr8`

A common application of calibration models in life sciences are enzymatic assays, where the quantification of glucose is one out of many popular examples. In this section, data from a glucose assay is used as a demonstration case for building calibration models with `calibr8`. First, the linear range of the assay is described by the corresponding linear calibration model to then explore an extended concentration range by implementing a calibration model with logistic trend of the location parameter. We examine a second calibration example that is nonlinear in its nature, namely the backscatter/biomass relationship of a commercially available cultivation device with online measurement, a BioLector Pro device (Section 3.1.1). Finally, we demonstrate how uncertainty estimates for biomass concentrations can be easily obtained with `calibr8`.

**4.1.1 Linear calibration model.** To acquire data for the establishment of a calibration model, 96 glucose standards between 0.001 and $50\frac{g}{L}$ were subjected to the glucose assay. A frequent approach to calibration modeling in life sciences is to identify the linear range of an assay and to discard measurements outside this range. From a previous adaptation of the glucose assay for automation with liquid handling robotics, the linear range was expected to be up to $2\frac{g}{L}$ (Holger Morschett, personal communication, 2019). Since samples are diluted by a factor of 10 before the assay, 83 glucose standards with concentrations below $20\frac{g}{L}$ remain for a linear calibration model.

As described in Section 2.4, calibration models use a probability distribution to describe the relation between independent variable and measurement outcome, both subject to random noise. In this example, we chose a Student-*t* distribution, thus the change of location parameter $\mu$ over the independent variable determines the trend of the calibration model. `calibr8` provides a convenience class `BasePolynomialModelT` that was used to implement a glucose calibration model with linear trend (Box 1 Code 1). For the spread parameter `scale`, we also chose a linear function dependent on $\mu$ to account for increasing random noise in dependency of the absorbance readout of the assay. Both can easily be adapted by changing the respective parameters `mu_degree` and `scale_degree` passed to the constructor of the convenience class. The degree of freedom $\nu$ in a `BasePolynomialModelT` is estimated from the data as a constant. The mathematical notation of this model can be found in S2 File.

The calibration model resulting from MLE of location and spread parameters was plotted with another `calibr8` convenience function (Fig 8A–8C). The plot shows the calibration model and measurement data (Fig 8A), the same relation with a logarithmic x-axis (Fig 8B)

and the relative residuals of data and model predictions (Fig 8C). As it is often recommended for biological assays, the glucose concentrations of the dilution series were evenly spaced on a logarithmic scale [60, 61], thus ensuring a higher accuracy of the model in the low-concentrated area (Fig 8B). To evaluate the quality of the linear calibration model, the residuals of data and model prediction were analyzed (Fig 8C). Overall, the residuals lie between ±5% of the observed values, demonstrating the high precision of the data. For concentrations higher than $0.6\frac{g}{L}$, an s-shaped trend is observed in the residuals, meaning that data first lies below and then above the linear model prediction. This indicates a lack-of-fit as described in Section 2.4. However, the discrepancy might also be caused by errors in the serial dilution that was pipetted with the robotic liquid handler, resulting in deviations from the expected linear relation. Moreover, it can be seen that the relative spread of residuals is quite constant, meaning that the absolute deviation increases with higher concentrations (Fig 8C). Although the linearly increasing `scale` parameter accounts for this rise of the underlying random noise, it can be

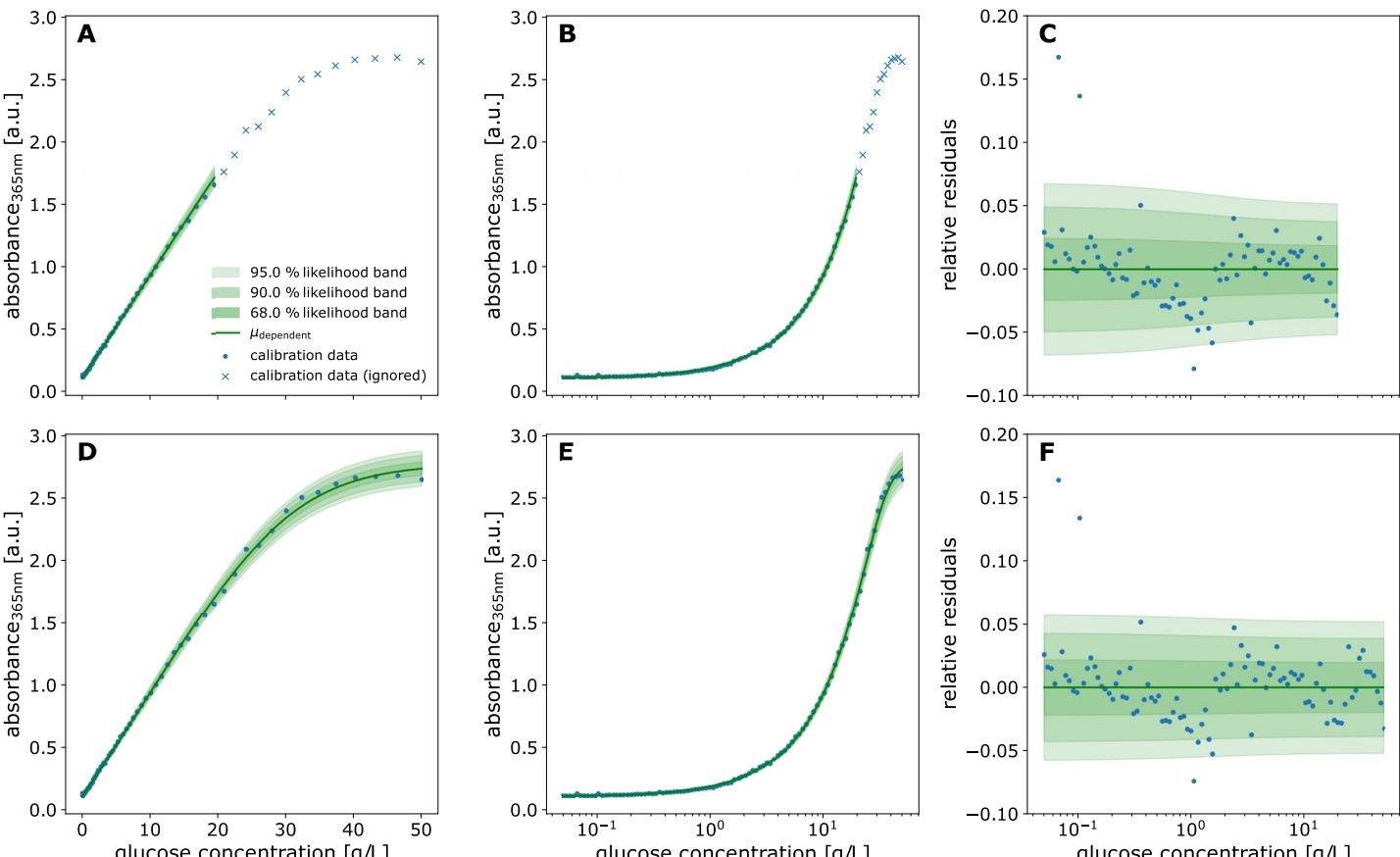

**Fig 8. Linear (top) and logistic (bottom) calibration model of glucose assay.** A calibration model comprising linear functions for both the location parameter $\mu_{A365}$ and the scale parameter of a Student-$t$ distribution was fitted to calibration data of glucose standard concentrations $(0.05 - 20 \frac{g}{L})$ and absorbance readouts by maximum likelihood estimation (**A-C**). The calibration data used to fit the linear model is the $0.05 - 20 \frac{g}{L}$ subset of standards that were spaced evenly on a log-scale up to $50 \frac{g}{L}$ (**B, E**). Likewise, a calibration model with a 5-parameter asymmetric logistic function for the $\mu$ parameter of the Student-$t$ distribution was fitted to the full $0.05 - 50 \frac{g}{L}$ calibration dataset (**D-E**). In both models, the scale parameter was modeled as a 1st-order polynomial function of $\mu$ and the degree of freedom $\nu$ as a constant. The extended range of calibration standard concentrations up to $50 \frac{g}{L}$ reveals a saturation kinetic of the glucose assay (**A, D**) and depending on the glucose concentration, the residuals (**C, F**) with respect to the modeled location parameter are scattered by approximately 5%. Modeling the scale parameter of the distribution as a 1st-order polynomial function of $\mu$ describes the broadening of the distribution at higher concentrations (**C**).

seen that it is slightly overestimated by the model since all data points above $2\frac{g}{L}$ lie within a 90% probability interval.

In comparison to simple linear regression, which is often evaluated by the coefficient of determination $R^2$ alone, the demonstrated diagnostics allow to judge whether the choice of model is appropriate. In this case, a more sophisticated model for the spread of the Student-$t$ distribution could be chosen to reduce the lack-of-fit. Moreover, all data points lying above $20\frac{g}{L}$ were not considered so far to allow for a linear model. In the following, we will therefore modify the existing calibration model to include a logistic function for the location parameter.

**4.1.2 Logistic calibration model.**   Although linear calibration models are useful in many cases, some relations in datasets are non-linear in their nature. Moreover, restricting analytical measurements to an approximately linear range instead of calibrating all concentrations of interest can be limiting. If the order of magnitude of sample concentrations is unknown, this leads to laborious dilution series or repetition of measurements to ensure that the linear range is met. In contrast, non-linear calibration models allow to describe complex relationships and, in case of biological assays, to reduce these time- and material-consuming workflows.

Many recommendations for experimental design in calibration can be found in literature (e.g. [60]). Having determined the range of interest for the calibration model, it should be exceeded in both directions if possible, thus ensuring that the relevant concentrations are well-covered. This way, all model parameters, including limits where applicable, can be identified from the observed data. Afterwards, the expected relationship between dependent and independent variable is to be considered. Since the glucose assay readout is based on absorbance in a plate reader (Section 3.1.4), which has a lower and upper detection limit, a saturation effect at high glucose concentrations is expected. In our demonstration example, glucose concentrations of up to $50\frac{g}{L}$ were targeted to cover relevant concentration for cultivation (Section 4.2) and at the same time to exceed the linear range towards the upper detection limit.

Sigmoidal shapes in calibration data, e.g. often observed for immunoassays, can be well-described by logistic functions [39]. In the `calibr8` package, a generalized logistic function with five parameters is used in an interpretable form (Section 3.2.1). It was used to implement a calibration model where the location parameter $\mu$ is described by a logistic function dependent on the glucose concentration. A respective base class `BaseAsymmetricLogisticT` is provided by `calibr8` (S1 File). The mathematical notation of the resulting model is given in S2 File. Using the whole glucose dataset up to $50\frac{g}{L}$, parameters of the new calibration model were estimated (Fig 8D–8F).

Overall, the logistic trend of the location parameter matches the extended calibration data well (Fig 8D and 8E). Since the scale parameter of the Student-$t$ distribution is modeled as a linear function dependent on $\mu$, the width of the likelihood bands approaches a limit at high glucose concentrations (Fig 8F). For concentrations greater than $3\frac{g}{L}$, no residuals lie outside of the 90% probability interval, indicating that the distribution spread is overestimated as it was before. Importantly, a direct comparison between the two calibration models (Fig 8C and 8F) reveals a high similarity in the reduced range ($< 20\ \frac{g}{L}$). This demonstrates how a non-linear model extends the range of concentrations available for measurement and modeling while improving the quality of the fit. For the glucose assay, truncating to a linear range thus becomes obsolete.

While non-linear models were so far shown to be useful to extend the usable concentration range of an assay, other applications do not allow to linearly approximate a subrange of measurements. Such an example is the online detection of biomass in growth experiments, where the non-invasive backscatter measurement (Section 3.1.1) does not allow for dilution of the cell suspension during incubation. To model the distribution of backscatter observations as a function

of the underlying biomass concentration, a structure similar to the glucose calibration model was chosen (S2 File). In contrast, the location parameter $\mu$ was modeled by a polynomial function of the logarithmic cell dry weight (CDW). The final CDW/backscatter calibration model was implemented using the `calibr8.BaseLogIndependentAsymmetricLogisticT` convenience class (Box 2 Code 2).

Two independent experiments were conducted to obtain calibration data as described in Section 3.1.5. The model was fitted to pooled data using the `calibr8.fit_pygmo` convenience function. As shown in Fig 9, the model accurately describes the nonlinear correlation between biomass concentration and observed backscatter measurements in the cultivation device (Fig 9A and 9B). Non-linearity is particularly observed for biomass concentrations below $10\frac{g}{L}$ (Fig 9A). Moreover, the residual plot (Fig 9C) mainly shows a random distribution; solely residuals between 1 and $3\frac{g}{L}$ indicate a lack-of-fit. To assess the potential influence, the resulting uncertainty in estimated biomass concentrations has to be considered, which will be further discussed in Section 4.1.3. Overall, the chosen logistic calibration model describes the calibration data well and is thus useful to transform backscatter measurements from the BioLector device into interpretable quantitative biomass curves.

In summary, this section illustrated how calibration models can be built conveniently with `calibr8` and showed that the asymmetric logistic function is suitable to describe many relationships in biotechnological measurement systems. Note that the only requirement to estimate `CalibrationModel` parameters with `calibr8` is to provide a `numpy.ndarray` with data for the independent and dependent variable. The code is agnostic of bioprocess knowledge and can thus be transferred to other research fields without further adaptations.

Having demonstrated how concentration/readout relations can be described by different calibration models, a remaining question is how to apply those calibration models. An important use-case is to obtain an estimate of concentrations in unknown samples, where uncertainty quantification is a crucial step.

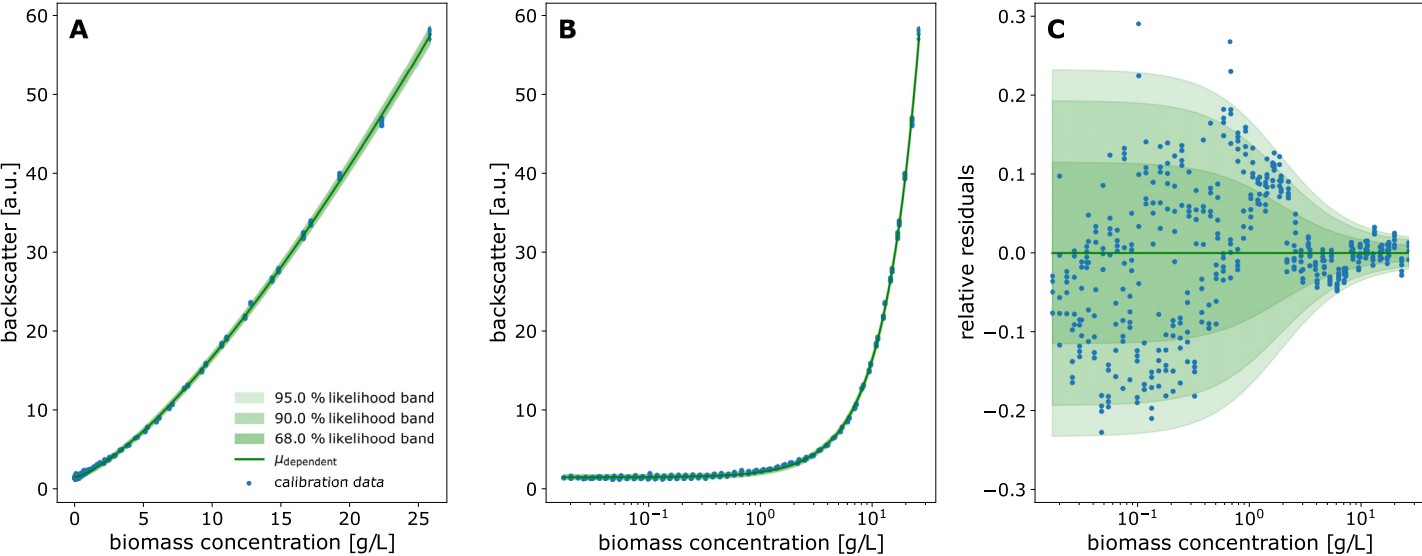

**Fig 9. Calibration model of biomass-dependent backscatter measurement.** Backscatter observations from two independent calibration experiments (1400 rpm, gain = 3) on the same BioLector Pro cultivation device were pooled. A non-linearity of the backscatter/CDW relationship is apparent already from the data itself (**A**). The evenly spaced calibration data (**B**) are well-described with little lack-of-fit error (**C**). At low biomass concentrations the relative spread of the measurement responses starts at ca. 20% and reduces to approximately 2% at concentrations above $10\frac{g}{L}$.

**4.1.3 Uncertainty quantification on independent variables.** After establishing a calibration model, the practitioner can in most cases consider the parameters of the model as fixed. Characterization of measurement reproducibility is thereby externalized into the calibration procedure, where random noise is inherently described by the spread of a probability distribution. The calibration model can then be put into application for the quantification of the independent variable from new observations. As introduced before, not only a single value of the independent variable is desired, but also a measure of uncertainty about it.

Quantifying the uncertainty in the independent variable as a continuous probability density is not only intuitive to visually interpret (Section 2.4), but also flexible with respect to the question of interest. To quantify the uncertainty numerically, various kinds of credible intervals (Section 2.3) can be obtained. For example, one might estimate the equal-tailed interval in which the independent variable lies with 90% probability, or alternatively the probability that it lies above a certain threshold.

In `calibr8`, the `CalibrationModel.infer_independent` method is used to perform the uncertainty quantification from one or more observations (Section 3.2.3). Internally, it uses the `loglikelihood` method of the calibration model and numerically integrates the sum of log-likelihoods over a user-specified range of plausible independent variables (Section 3.2.3). The resulting `calibr8.ContinuousUnivariateInference` is equivalent to Eq 13, where the prior $p(x)$ is specifying the plausible range.

$$p(x \mid \vec{y}_{obs}) \quad = \frac{\mathcal{L}(x \mid \vec{y}_{obs}) \cdot p(x)}{\int_{-\infty}^{\infty} \mathcal{L}(x \mid \vec{y}_{obs}) \cdot p(x) \ dx} = \frac{\mathcal{L}(x \mid \vec{y}_{obs}) \cdot p(x)}{\int_{a}^{b} \mathcal{L}(x \mid \vec{y}_{obs}) \ dx}$$

$$\text{where } p(x) \quad = \text{Uniform}(a, b)$$

(13)

For convenience, the `CalibrationModel.infer_independent` method automatically determines median and credible interval (ETI and HDI) bounds. It determines vectors for the independent variable and the conditional probability density that can be plotted without further processing.

In Fig 10, various inferences obtained with `infer_independent` are illustrated with a biomass calibration model. For illustration purposes, the calibration model from Section 4.1.2 was slightly modified and the degree of freedom $v$ was modified to an extreme value of 1. Different sets of simulated observations were subjected to the `infer_independent` method and the resulting credible intervals are shown in subplots A and B.

In Fig 10A, the orange curves show the 95% ETIs for one, two or three observations of back-scatter signals corresponding to $1.5\frac{g}{L}$. As expected, the probability mass is concentrated around $1.5\frac{g}{L}$ with higher number of observations and the ETIs get narrower. Furthermore, observations in the lower or upper saturation of the measurement system typically result in one-sided probability densities (blue).

When the calibration model assumes the possibility of outliers (Student-$t$ distributed measurement responses), the observation of drastically different measurement responses can translate into a multi-modal posterior belief in the independent variable. The intuition behind this multi-modality is that a subset of observations are "outliers" from the perspective of the remaining observations and vice versa. In the example shown in Fig 10B, the three observations around 0.5 could be "outliers", or the ones around 1.3, but from the data alone both are equally likely. Hence the posterior belief in the biomass concentration is bimodal. The green arrows indicate the MLE estimates obtained via `predict_independent` with individual observations. The plot also reveals slight differences between ETI and HDI in this scenario.

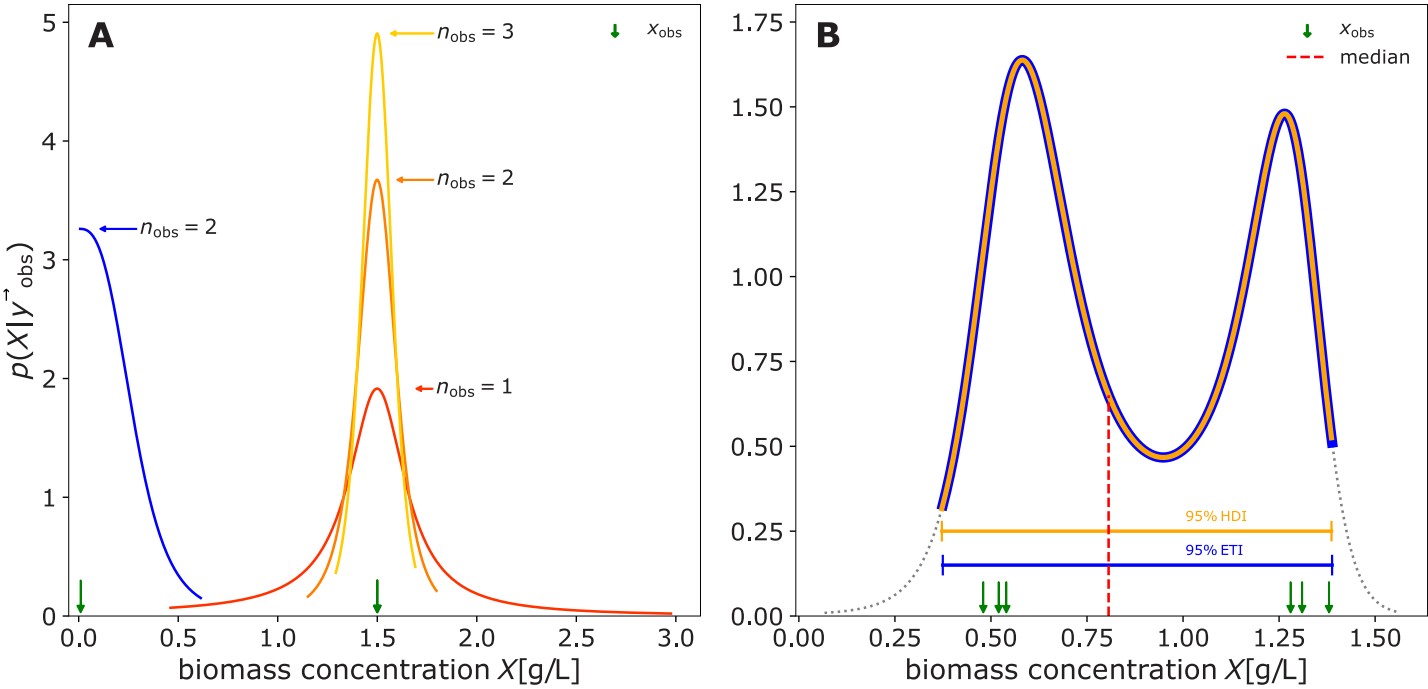

**Fig 10. Independent variable PDFs in various observation scenarios.** Posterior densities inferred from various numbers of observations corresponding to different biomass concentrations are shown (**A**). The ends of the drawn lines in **A** indicate the 95% equal-tailed interval. Near biomass concentrations of 0, the posterior density is asymmetric (**A**, blue), indicating that very low concentrations cannot be distinguished. As the number of observations grows, the probability mass is concentrated and the ETIs shrink (**A**, oranges). The choice of a Student-*t* distribution model can lead to a multi-modality of the inferred posterior density when observations lie far apart (**B**). For asymmetric distributions, the median (dashed line) does not necessarily coincide with a mode and equal-tailed and highest-density intervals (ETI, HDI) can be different. Maximum likelihood estimates from individual observations, as obtained via `predict_independent` are shown as arrows. Note: $\vec{y}_{obs}$ and the model's $v$ parameter were chosen at extreme values for illustrative purposes.

Note that both the degree of freedom and the simulated observations have been set to extreme values to illustrate the properties of the Student-*t* distribution.

The Bayesian or likelihood-based perspective on uncertainty in the independent variable (Eq 13) allows for quantification of uncertainty even with single observations, close to zero, or close to saturation limits of the measurement system. Calibration models built with `calibr8` are easy to set up, visualize and diagnose and can thus be flexibly integrated into existing data analysis workflows of various domains. Moreover, the setup in a versatile, object-oriented programming language such as Python allows to use `calibr8` in high-throughput, automated experiments where hundreds of calibration models must be fitted. Next, we will build upon the presented biomass and glucose calibration models and demonstrate how combining them with a bioprocess model enables to gain insight into the growth phenotype of a biotechnological model organism, *Corynebacterium glutamicum*.

## 4.2 Application 2: Process modeling of bacterial growth

A real-world experimental procedure is often not a textbook example but rather a heterogeneous dataset, e.g. comprising multiple measurement types or varying process conditions. We use the term process model, as introduced in Section 2.5, to describe the complete underlying chemical or biological process, but not the experimental observations that are made of it. These input/output relations of the measurement system are explicitly described by calibration

models. In this application example, we demonstrate how object-oriented `calibr8` calibration models from Section 4.1 can be combined with an ODE bioprocess model to describe a heterogeneous dataset of bacterial growth curves.

The simplest experimental setup to obtain a bacterial growth curve is a so-called batch cultivation. Under laboratory conditions, such batch cultivations can be performed in a variety of cultivation systems such as shake flasks, bioreactors or miniaturized reaction systems. From a data science perspective, the cultivation systems differ mostly by how many independent cultivations are performed in parallel and by the kind and number of observations made per cultivation. In the domain of bioprocess development, a large number of cultivations must be conducted to find best-performing producer strains and media compositions. For these applications, microbioreactors, which are miniaturized shaken or stirred cultivation systems, offer an increased cultivation throughput combined with non-invasive online measurements such as pH or dissolved oxygen tension (DO) [62]. In this study, a commercially available microbioreactor called BioLector Pro (Section 3.1.2) was used, which additionally provides quasi-continuous measurement of biomass. Note that for the understanding of the underlying software, the cultivation device is of minor importance and any other measurement device could have been used instead. All three signals (pH, DO and biomass) in this example are obtained optically and must be calibrated against the true variable of interest (Section 3.1.1, Section 3.1.5). Furthermore, confounding factors are known for all three measurement methods, mandating special rigor in the design and analysis of quantitative experiments. For example, the optode-based pH and DO measurements can be influenced by media components, or the backscatter signal by morphology changes.

To facilitate a simple application example, we grew *C. glutamicum* in parallel, miniaturized batch cultivations within a specialized deepwell plate (FlowerPlate) (Section 3.1.2). This bacterium is a well-known industrially applied microorganism that exhibits textbook-like exponential growth kinetics when grown on carbon sources such as glucose [63]. A preculture was grown in two wells (A01 and B01) of the deepwell plate and used to automatically inoculate 28 main culture wells (A02 through D08). We thus avoided a lag phase of adaptation at the beginning of the growth curve, which greatly simplifies the process model (Section 3.1.2). As we will see later on, the pipetting error of the robotic liquid handler at the small inoculation volume must be considered when setting up the process model, highlighting the need to adapt the data analysis to the peculiarities of the experiment.

Before going into the details of the process model for this application example, we would like to emphasize that the same modeling techniques can be applied to other domain-specific examples.

**4.2.1 Building an ODE process model for bacterial growth experiments.** The simplest model for microbial growth is the Monod kinetics differential equation model of substrate-limited exponential growth [64]. Similar to how the famous Michaelis-Menten kinetics describe enzymatic reaction rates, the Monod kinetics model the specific growth rate as a function of substrate concentration. Under the assumptions of homogeneous mixing, unlimited nutrient supply and constant ambient conditions, the Monod model can be applied to batch cultivations of bacterial, fungal, plant or cell cultures that grow with a maximum growth rate $\mu_{max}$ until a substrate, typically a carbon source, is depleted.

The Monod model (Eq 14) has five parameters including the initial conditions for substrate concentration $S_0$ and biomass concentration $X_0$. The maximum growth rate $\mu_{max}$ specifies the specific exponential growth rate that the organism can achieve under the modeled conditions. The actual specific growth rate $\mu(t)$ is modeled as a function of $\mu_{max}$, the current substrate concentration $S$ and a parameter $K_S$ that corresponds to the substrate concentration at which

$\mu(t) = \frac{\mu_{max}}{2}$. The last parameter $Y_{XS}$, called biomass yield, describes the amount of substrate consumed per unit of formed biomass.

$$\begin{aligned}
\frac{dX}{dt} &= \mu_{max} \cdot X \cdot \frac{S}{K_S + S} \\
\frac{dS}{dt} &= -1/Y_{XS} \cdot \frac{dX}{dt}
\end{aligned}$$
$$S, X, \mu_{max}, K_S, Y_{XS} \in \mathcal{R}_{>0}$$

(14)

The experiment to be modeled in this application example was devised such that Monod-like growth behavior of *C. glutamicum* wild-type could be expected (Section 3.1.2). We grew 28 parallel batch cultures that were sampled to measure glucose concentrations in addition to the high-resolution backscatter time series. The resulting dataset comprises 28 replicates, each with backscatter time series of varying length and a time series of length 1 for the glucose absorbance readout. Building upon our Python package `murefi` for flexible **mu**lti-**re**plicate **fi**tting, we loaded the raw observations into a `murefi.Dataset` object (Section 3.2.4). The package was designed to simplify the definition and parameter estimation of process models that describe all replicates in a dataset simultaneously.

To build such elaborate process models with `murefi`, the user must specify the process model corresponding to a single replicate, as well as a set of rules that describe how parameters of this model are shared across replicates. The Monod kinetics in this application example were implemented in just a few lines of code by subclassing from `murefi.BaseODEModel` (Box 3 Code 3).

For heterogeneous datasets, the rules for sharing process model parameters across replicates can be complex and hard to implement and most modeling workflows require the practitioner to often change the parametrization. In `murefi`, the `ParameterMapping` class supports the modeler by specializing in the tedious translation of parameter sharing rules into a function (`.repmap(...)`) that takes a single parameter vector and transforms it into replicate-specific parameter vectors. At the same time, it provides mechanisms for specifying fixed parameters, initial guesses and bounds on the parameters. Reading a spread sheet with parameters into Python is an easy way of initializing the `ParameterMapping` (Table 1).

Unique names specify that a parameter is only estimated from the indicated replicate (e.g. X0_A02) while shared names correspond to global parameters (e.g. S0). For the application example at hand, a parameter mapping was defined such that the parameter $X_0$ is local to

**Table 1. Tabular representation of a parameter mapping.** With columns corresponding to the parameter names of a naive Monod process model, the parametrization of each replicate, identified by a replicate ID (rid) is specified in a tabular format. Parameter identifiers that appear multiple times (e.g. S0) correspond to a parameter shared across replicates. Accordingly, replicate-local parameters names simply do not appear multiple times (e.g. X0_A06). Numeric entries are interpreted as fixed values and will be left out of parameter estimation. Columns do not need to be homogeneously fixed/shared/local, but parameters can only be shared within the same column. The parameter mapping can be provided as a `DataFrame` object.

| rid | $S_0$ | $X_0$ | $\mu_{max}$ | $K_S$ | $Y_{XS}$ |
|-----|-------|-------|-------------|-------|----------|
| A02 | S0 | X0_A02 | mu_max | 0.02 | Y_XS |
| A03 | S0 | X0_A03 | mu_max | 0.02 | Y_XS |
| A04 | S0 | X0_A04 | mu_max | 0.02 | Y_XS |
| A05 | S0 | X0_A05 | mu_max | 0.02 | Y_XS |
| A06 | S0 | X0_A06 | mu_max | 0.02 | Y_XS |

Box 4. Code 4. MLE of process model parameters.

```
1   model = MonodModel()
2   dataset = murefi.load_dataset("cultivation_dataset.h5")
3   cm_biomass = BioLectorCDWBackscatterModelV1.load("biomass_cm_logistic.json")
4   cm_glucose = LogisticGlucoseCalibrationModelV1.load("glucose_cm_logistic.json")
5   df_mapping = pandas.read_excel("parameter_mapping.xlsx", index_col="rid")
6
7   theta_mapping = murefi.ParameterMapping(
8       df_mapping,
9       bounds={
10          "S0": (15, 20),
11          "X0": (0.01, 0.4),
12          "mu_max": (0.4, 0.5),
13          "Y_XS": (0.3, 1)
14      },
15      guesses={
16          "S0": 17,
17          "X0": 0.01,
18          "mu_max": 0.4,
19          "Y_XS": 0.5
20      }
21  )
22  objective = murefi.objectives.for_dataset(
23      dataset=dataset,
24      model=model,
25      parameter_mapping=theta_mapping,
26      calibration_models=[cm_glucose, cm_biomass]
27  )
28  mle_result = scipy.optimize.minimize(
29      objective,
30      x0=theta_mapping.guesses,
31      bounds=theta_mapping.bounds
32  )
```

each replicate while $S_0$, $\mu_{max}$ and $Y_{XS}$ are shared across all replicates. For the Monod substrate affinity constant $K_S$, literature reports values of approximately $0.0005 - 0.1 \frac{g}{L}$ for *Escherichia coli* [65]), while no data is available for *C. glutamicum*. Because it is practically non-identifiable at the resolution of our dataset, $K_S$ was fixed to an arbitrary, but numerically harmless value of $0.02 \frac{g}{L}$. In Table 1, this is expressed by the numerical column entries.

A likelihood function for parameter estimation was created using the `murefi.objectives.for_dataset` convenience function (Box 4 Code 4). The objective is independent of the parameter estimation paradigm and was applied for optimization via MLE (Section 4.2.2) and sampling by MCMC (Section 4.2.3) in the scope of this work.

**4.2.2 Estimating ODE process model parameters by maximum likelihood.** First, we determined maximum likelihood estimates of the process model parameters through optimization. In few lines of code, the calibration models from Section 4.1 and dataset are loaded (Box 4 Code 4, ll. 2–4), the process model is instantiated (Box 4 Code 4, l. 1) and the

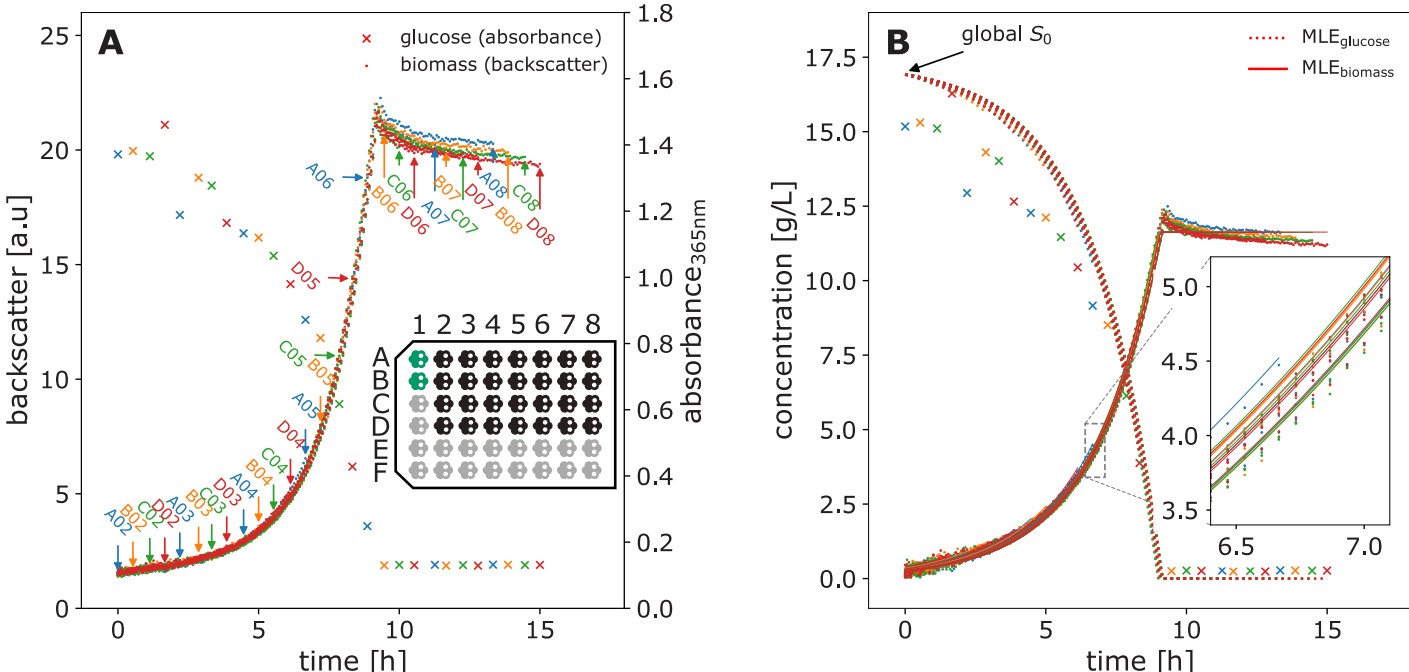

**Fig 11. Measurements and maximum likelihood estimate of *C. glutamicum* growth Monod model.** Original measurement responses of online biomass (backscatter) and *at-line* endpoint glucose assay measurements (absorbance) are shown in (**A**). Glucose measurements were obtained by sacrificing culture wells, hence each backscatter time series terminates at the time of glucose assay observations. The time and well ID of sacrifices are marked by arrows, colored by row in the cultivation MTP. The inset plot shows a typical layout of the cultivation plate (FlowerPlate). The preculture wells are highlighted in green, main cultures in black. In **B**, the observations and MLE predictions of the ODE process model are shown in SI units. Observations were transformed from original units using the `predict_independent` method of the respective calibration model. Whereas all curves start at the same global initial substrate concentration $S_0$, each well has individual initial biomass concentrations, resulting in the time shifts visible in the zoomed-in inset plot. Biomass observations in the inset plot (●) correspond to the median posterior inferred from each backscatter observation individually.

`ParameterMapping` is specified with bounds and guesses (Box 4 Code 4, ll. 7–21). The `objective` (Box 4 Code 4, ll. 22–27) can directly be used for an optimization algorithm (Box 4 Code 4, ll. 28–32), in this case one from the popular Python library `scipy`. A table with MLE parameters can be found in S1 Table.

Fig 11 shows the observations alone (A) and combined with MLE results (B) for glucose (absorbance) and biomass (backscatter). The replicates were sampled at different times to measure glucose concentrations; the end of a time series is indicated by an arrow and the replicate name (Fig 11A). Overall, the backscatter time series show a very high reproducibility, which demonstrates the effect of pooling precultures before inoculation (Section 3.1.2). The model describes the observations so accurately that they can only be distinguished in the inset plot (Fig 11B). Here, a small difference between replicates can be observed, which is caused by varying initial biomass concentrations due to inevitable pipetting errors in the automated inoculation of the main cultures. It becomes evident that replicate-wise $X_0$ parameters were necessary to account for this effect. The different initial biomasses are also visible from the spread of data points at the beginning of the growth curve (Fig 11B). For the biomass, the only period of systematic deviation between model prediction and observations is at the time of entry into the stationary phase, the phase where substrate is depleted and growth stops. Here, the biomass signal overshoots while the Monod kinetics predict a rapid change to a constant signal. This effect in the growth curve of *C. glutamicum* is also known

from other experiments with the BioLector [66] and cannot be accounted for by the otherwise useful textbook process model.

The glucose data shows more deviation, but follows the expected textbook behaviour of exponential decay (Fig 11B). Interestingly, the predictions for glucose concentrations at the end of cultivation lie slightly above $0 \frac{g}{L}$, showing that the corresponding calibration model is not describing this range of concentrations well. The deviation could be caused by other components in the used cultivation medium that distort the measurement compared to calibration with fresh medium as diluent. However, this was not further investigated since the substrate data has little influence on the parameter estimation compared to the high-resolution back-scatter measurements.

From a first inspection of MLE results, we can see that the simple Monod process model describes the high-resolution data very well. For more insight, we will take a look at the parameter estimation, correlations and systematic deviations using a Bayesian approach.

**4.2.3 Hierarchical Bayesian ODE models with `calibr8` and `murefi`.** The results presented in the previous chapter show that the Monod model, when combined with non-linear calibration models for the observations, can describe the observed biological process with high accuracy. However, the precision of the parameter set obtained by the maximum likelihood method is still unknown. Particularly, when decisions are made from model-based inferences and predictions, the uncertainty about these variables is a key factor.

The combination of (forward) sensitivity analysis with Gaussian error propagation could be applied to learn about the precision of the maximum likelihood estimate. Instead of maximum likelihood optimization of a parameter set, Bayes' rule can be used to infer a posterior probability distribution of parameters. In comparison to the maximum likelihood method, the Bayesian approach allows to incorporate prior knowledge and inherently quantifies uncertainty and parameter correlations. Bayesian posteriors can in some (rare) cases be obtained analytically, or numerically as shown in Section 4.1.3. However, in most practical applications Markov chain Monte Carlo (MCMC) algorithms are applied. MCMC offers convergence guarantees as the number of iterations approaches infinity and can give satisfactory results with competitive computational performance when modern algorithms are used.

To build a Bayesian process model, one must explicitly state prior beliefs in the model parameters in the form of probability distributions. For our hierarchical Monod model application example, we must specify prior beliefs in the ODE parameters $\mu_{\max}$, $Y_{XS}$ and initial conditions $S_0$ and $X_{0,\text{well}}$. Prior distributions for these parameters were specified to reflect biologically reasonable, but uninformative assumptions about the experiment (Eq 15). The initial substrate concentration $S_0$ was expected at approximately $20 \frac{g}{L}$ with a conservative 10% relative error. For *C. glutamicum* wild-type, our priors for biomass yields with $\text{HDI}_{Y_{XS}}^{95\ \%} = [0.5, \ 0.7] \frac{g_{\text{CDW}}}{g_{\text{glucose}}}$ and for maximum growth rates with $\text{HDI}_{\mu_{\max}}^{95\ \%} = [0.2, \ 0.6] \ h^{-1}$ are uninformative and based on literature [67]. Our process model describes initial biomass concentrations on a per-well basis (Section 4.2.1), but can still infer the mean initial biomass concentration $X_{0,\mu}$ as a hyperprior by modeling well-specific offsets w.r.t. the group mean as $X_{0,well} = X_{0,\mu} \cdot F_{\text{offset,well}}$. Through $X_{0,\mu}$ the priors for all initial biomass concentrations $\vec{X}_0$ are parametrized by a common parameter, allowing each individual $X_{0,\text{well}}$ to vary while concentrating around their common group mean. For more intuition and details about Bayesian hierarchical modeling in particular, we refer to [68]. While the experiment performed here is modeled with a hierarchical prior on a process model parameter, one may also use hierarchical priors for calibration model parameters. A corresponding example is shown in the `calibr8` documentation [22].

The experiment was programmed to inoculate main cultures to approximately $0.25 \frac{g}{L}$ (Section 3.1.2), therefore the prior for $X_{0,\mu}$ was centered at $0.25 \frac{g}{L}$ with a 10% relative error. Our prior belief in the well-specific relative offset $F_{\text{offset,well}}$ was also modeled by a Lognormal distribution with mean 0, corresponding to the expectation that the offset is centered around 1 in the transformed space. A standard deviation of 20% was chosen to account for random and systematic inaccuracy of the automated liquid handler at the low pipetting volume of 20 μL [66].

$$
\begin{aligned}
X_{0,\mu} &\sim \text{Lognormal}(\mu = \log(0.25), \sigma = 0.1) \\
\vec{F}_{\text{offset}} &\sim \text{Lognormal}(\mu = 0, \sigma = 0.2) \\
\vec{X}_0 &\sim X_{0,\mu} \cdot \vec{F}_{\text{offset}} \\
\\
S_0 &\sim \text{Lognormal}(\mu = \log(20), \ \sigma = 0.1) \\
Y_{\text{XS}} &\sim \text{Beta}(\mu = 0.6, \ \sigma = 0.05) \\
\mu_{\text{max}} &\sim \text{Beta}(\mu = 0.4, \ \sigma = 0.1)
\end{aligned}
\tag{15}
$$

$$
\begin{aligned}
Y_{\text{pred, well}} &\sim \phi_{\text{process model}}(S_0, \ X_{0,\text{well}}, \ \mu_{\text{max}}, \ Y_{\text{XS}}) \\
\mathcal{L}(\theta_{\text{pm}} \mid Y_{\text{obs}}) &= p(Y_{\text{obs}} \mid \phi_{\text{calibration model}}(Y_{\text{pred}}))
\end{aligned}
$$

When modeling with `calibr8` and `murefi`, this specification of prior beliefs is the only overhead compared to the MLE method. The API of both packages was designed to be fully compatible with the probabilistic programming library PyMC, such that `calibr8` and `murefi` models can become fully Bayesian with little programming effort.

Concretely, the objective function created by `murefi` accepts Aesara tensors (e.g. PyMC random variables) as inputs, resulting in a symbolic `TensorVariable` likelihood instead of a numeric one. The PyMC model for the hierarchical ODE process model in our application example builds upon the previously established `objective` function (Box 4 Code 4, l. 22). The model code (Box 5 Code 5) resembles the mathematical notation of the same model shown in Eq 15.

After the PyMC process model was defined, its parameters were estimated by MCMC as described in Section 3.2.6. Two-dimensional marginals of the posterior samples obtained from MCMC sampling are shown in Fig 12 for two replicates and in S2 Fig for the whole dataset.

The pair plot visualization of the posterior reveals that some model parameters are strongly correlated with each other. Among those strong correlations are the pair of initial substrate concentration $S_0$ and biomass yield $Y_{\text{XS}}$. Interestingly, correlations were found even in the very narrow HDIs of $X_{0,\text{well}}$ and $\mu_{\text{max}}$, which is particularly clear for replicate D06. An interpretation is that when the initial biomass concentration is estimated at a smaller value, the maximum growth rate of cells must be higher to reach the same biomass level. The correlation is thus a natural consequence of the underlying process. Similarly, a lower initial substrate concentration results in a higher yield.

From a modeling point of view, the plot reveals how identifiable the model parameters are from the data. Furthermore, strong correlations, as observed for $Y_{\text{XS}}$ and $S_0$, can be problematic for some optimization or MCMC sampling algorithms. In this case, the applied algorithm DE-Metropolis-Z [51] proved beneficial to sample the 32-dimensional parameter space with highly correlated parameters (Fig 12, top right). Interestingly, the strength of the correlation depends on the amount of data that was available for a particular replicate (S1 File). The more

Box 5. Code 5. Specification of complete process model in PyMC.

```
with pymc.Model() as pmodel:
    # Specify a hyperprior on the initial biomass group mean:
    # + centered on the planned inoculation density (0.25 g/L) in main cultures
    # + with a 10 % standard deviation to account for pipetting errors
    X0_mu = pymc.Lognormal("X0_mu", mu=numpy.log(0.25), sd=0.10)

    # Model the relative offset of initial biomass between each well and
    # the group mean with a relative pipetting error of 20 %
    F_offset = pymc.Lognormal("F_offset", mu=0, sd=0.20, shape=(N_wells,))

    # Thereby, the initial biomass in each well is the product
    # of group mean and relative offset:
    X0 = pymc.Deterministic("X0", X0_mu * F_offset)

    # Combine the priors into a dictionary
    theta = {
        "S0": pymc.LogNormal("S0", mu=numpy.log(20), sigma=0.10),
        "Y_XS": pymc.Beta("Y_XS", mu=0.6, sd=0.05),
        "mu_max": pymc.Beta("mu_max", mu=0.4, sd=0.1),
        # unpack the vector of initial biomasses into individual scalars
        **{
            f"X0_{well}": X0[w]
            for w, well in enumerate(wells)
        }
    }
    # Re-use the objective function from the MLE model code
    L = objective(theta)
```

data available, the stronger the correlation between $X_0$ and $\mu_{\max}$; this can also be observed for wells D04 and D06. The parameter estimates by MCMC are also tabulated in S1 Table.

In the lower part of Fig 12, the observations as well as model predictions with parameters sampled from the posterior are shown. Each line in the density plot corresponds to one set of parameters sampled with the MCMC algorithm. The small width of the density bands express how well the parameters could be estimated from the data, which is in accordance to the pair plot above. The violins around the substrate data visualize the uncertainty of glucose concentration inferred with the calibration model alone, instead of using the process model with all evidence. The violin is wider than the posterior band from the process model accordingly. Similar to the the MLE results, it becomes obvious that the Monod model estimate is well-suited to describe the biological dataset. With `calibr8` and `murefi`, building and sampling the Bayesian model needs a similar effort as MLE and the user can focus on structural requirements rather than cumbersome implementation. To assess the benefits of the Bayesian model in more detail, the role of different calibration models, the residuals and the hierarchical parameter $X_0$ are investigated in more detail in the next section.

**4.2.4 Process and model insight through Bayesian uncertainty quantification.** From the process model fit and the uncertainty estimates in particular, one can draw conclusions about the choice of model and the underlying biological process. First, to emphasize that the

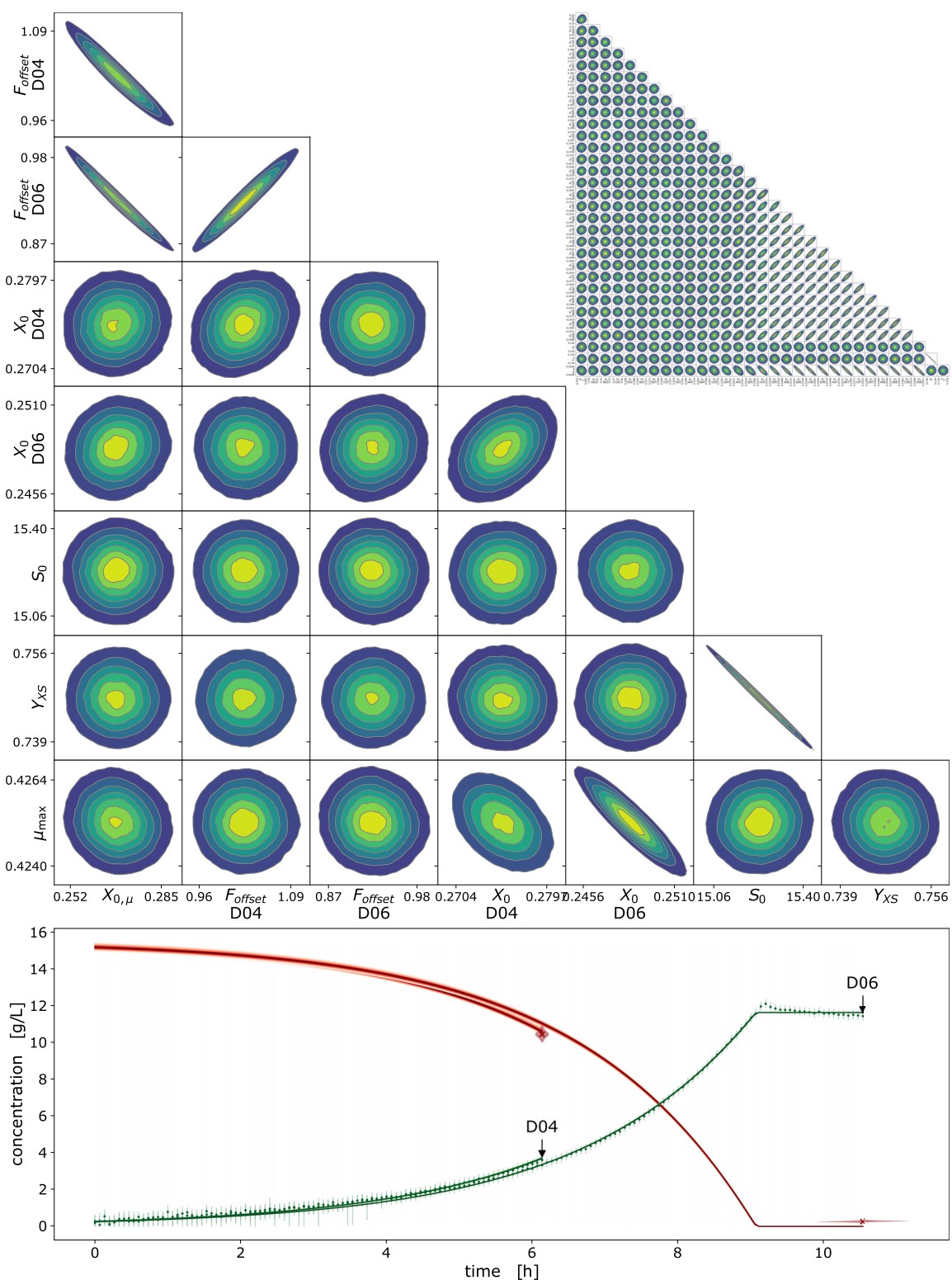

**Fig 12. Parameter correlations, data and posterior distributions of growth curves.** Each kernel density estimate (KDE) in the top half shows a 2-dimensional cross section of the full posterior, visualizing correlations between some of the model parameters. For example, the topmost KDE shows that the posterior samples of $F_{\text{offset,D04}}$ are correlated with $X_{0,\mu}$. Axis labels correspond to the lower and upper bound of 90% HDIs. The large pair plot shows just the marginals that are relevant for the replicates D04 and D06, whereas the small pair plot shows the dimensions for all parameters (high resolution in S2 Fig). In the bottom half of the figure, the kinetics of

replicates D04 and D06 are drawn. The red (substrate) and green (biomass) densities correspond to the distribution of predictions obtained from posterior samples, as described in Section 3.2.7. The red violins visualize the posterior inferred from single glucose measurement responses without the use of the process model. Likewise, the green vertical bars on the biomass concentrations show the 90% HDI.

elaborate non-linear calibration model was required, we compare the process model fits obtained with a non-linear versus a linear calibration model. The more traditional linear biomass/backscatter correlation was fitted to calibration data as described in Section 3.1.5 and used to fit the D06 replicate from our dataset. For comparison, the asymmetric logistic calibration model from Section 4.1.1 was used to estimate parameters of the same process model and data.

On a first glance, the fit of the Monod process model using the linear biomass calibration model looks like a good description of the data (Fig 13A), but does not hold up to closer inspection. The residual plots (Fig 13B and 13C) reveal that using the linear calibration model results in systematically larger residuals of the process model compared to using the logistic calibration model. A thorough inspection of the linear calibration model itself (Fig 13D) also reveals that it already has a lack-of-fit of the location parameter (green line), similar to the depiction in Fig 3. We would like to point out that also the maximum growth rate estimated from a process model with linear biomass/backscatter calibration ($\mathrm{HDI}^{90\,\%}_{\mu_{\max}} = [0.480,\ 0.530]$) is systematically overestimated compared to the one with the logistic model ($\mathrm{HDI}^{90\,\%}_{\mu_{\max}} = [0.414,\ 0.423]$). Regarding the choice of calibration model for the biomass/backscatter relationship, we conclude that the linear model should no longer be used, as it results in biased parameter estimates.

Having chosen a suitable calibration model for the variables, the choice of the Monod model itself can be investigated. Fig 14 shows the high-resolution biomass data and predictions from MCMC on a logarithmic y-scale (Fig 14A) as well as the residuals in backscatter units

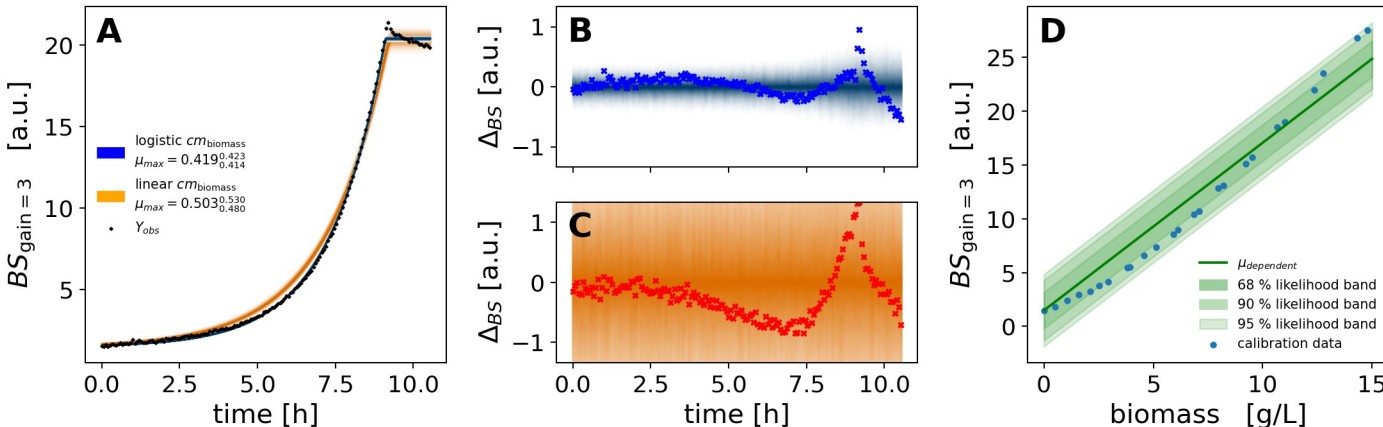

**Fig 13. Comparison of Monod model fit with linear error model.** Two Monod kinetic process models were fitted to the same observations from culture well D06 utilizing either a linear calibration model for the biomass/backscatter relationship (orange in **A**, calibration in **D**) or the previously established logistic model (blue in **A**). In **A** the posterior distribution of backscatter observations (density bands) is overlaid with actual backscatter observations. A linear calibration (**D**) model with fixed intercept (Section 3.1.5) was fitted to the subset of calibration data points up to $15\frac{g}{L}$ such that it covers the range of biomass concentrations expected in the experiment. Residual plots of the observations compared to the posterior predictive distribution of backscatter observations (**B, C**) show that the fit obtained with the logistic calibration model (blue) has much less lack-of-fit compared to the one with the linear model (orange). Note that the backscatter residuals of ±1% are small compared to the amplitude of the absolute values going from close to 0 to approximately 20. The discrepancy between the two models is also evident from the 90% HDI of the maximum growth rate $\mu_{max}$ of [0.414, 0.423] $h^{-1}$ in the logistic and [0.480, 0.530] $h^{-1}$ in the linear case.

(Fig 14B). In the left subplot, the data was transformed to biomass concentrations with the logistic biomass calibration model. The orange intervals represent the $\text{HDI}^{90\ \%}_{\text{biomass}}$ inferred from a single observation using only the calibration model. In contrast, the blue density represents the posterior of the process model, which contains all observations. Naturally, the posterior from all evidence, combined through the process model, is much narrower than the posterior from any single observation. The plot reveals that the exponential growth assumed by the Monod model is generally suitable for the growth on glucose, since the blue density is describing the trend of observations well.

To evaluate a lack-of-fit, the residual plot (Fig 14B) should be considered. Here, the residuals between the process model posterior and the observed backscatter are shown in black, the respective posterior predictive distribution of measurement responses (Section 3.2.3) is shown in green. The posterior predictive is the distribution of measurement responses that the model predicts. First, biomass concentrations are drawn from the posterior distribution. At each biomass concentration, another sample is taken from the Student-$t$ distribution predicted by the biomass calibration model.

A large deviation that cannot be explained with the uncertainty of the estimate can be observed after 8 hours. Looking at the data, e.g. in Section 4.2.4, it can be seen that it accounts for the previously described overshoot of the backscatter signal at the beginning of the stationary phase (Section 4.2.2). This phenomenon cannot be explained by the Monod model, which assumes a constant biomass concentration after substrate depletion. Further investigations are needed to identify whether the change is morphological, e.g. a shrinking of cells due to carbon source depletion, or a decrease of biomass, e.g. by cell lysis.

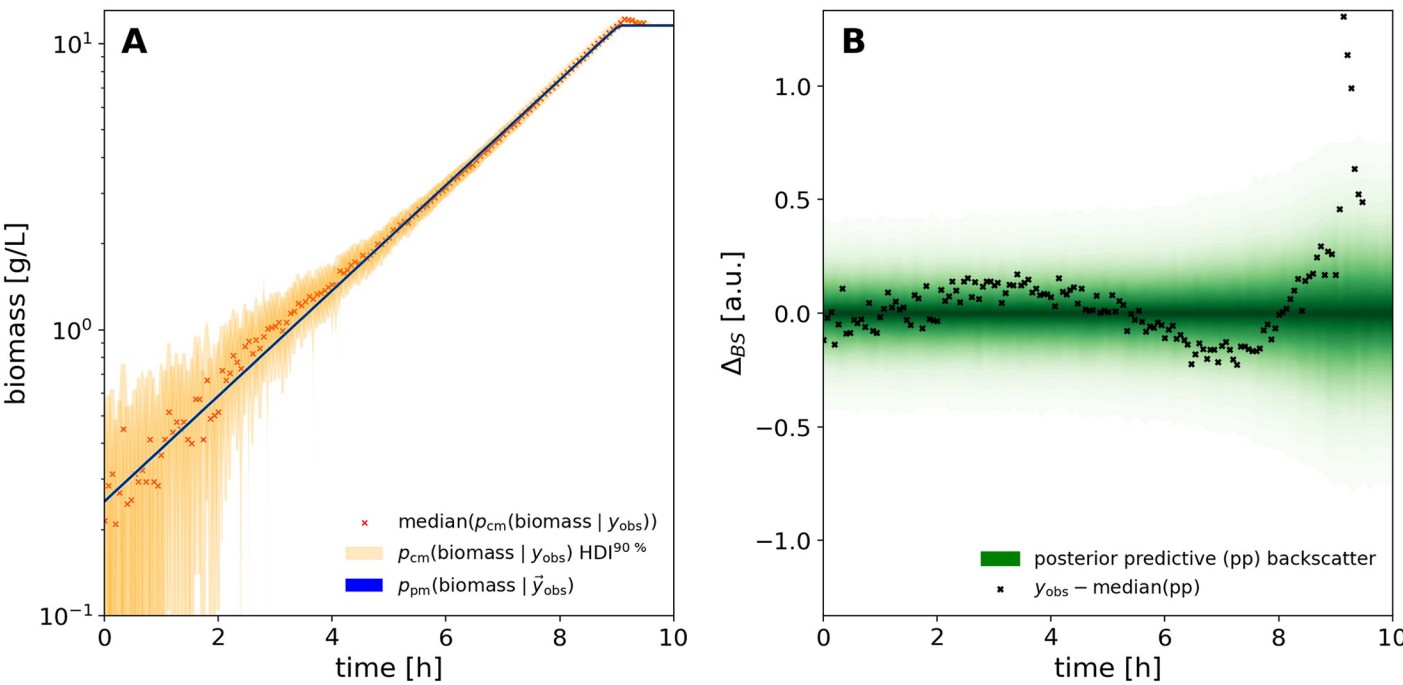

**Fig 14. Predictions, observations and residuals of Monod model fitted to backscatter data. A**: Through a logarithmic y-axis, the plot **A** shows that both process model (blue density) and the $\text{HDI}^{90\ \%}_{\text{biomass}}$ obtained from the biomass calibration model with individual observations (orange) describe an exponentially increasing biomass concentration up to approximately 9 hours. **B**: The residuals between prediction and observed backscatter (black) and the posterior predictive backscatter distribution (green density) show that the lack-of-fit is consistently less than ±0.25 backscatter units with the exception of a fluctuation at the time of substrate depletion.

Before 8 hours, an s-shaped systematic deviation can be observed, meaning that the observations first lie above and then below the prediction. Apart from the influence of the overshoot, which distorts the fit, this might be explained by a different growth rate. It was previously shown that *C. glutamicum* exhibits a higher specific growth rate on protocatechuic acid (PCA), which is a component of the cultivation medium CGXII [67]. Upon depletion of PCA after the first hours of cultivation, the growth rate decreases accordingly. This is not accounted for in the Monod kinetics, which describe an effectively constant growth rate at substrate concentrations much higher than the $K_S$ value. To cover this effect, PCA must be measured, e.g. by sampling and liquid chromatography, and a more elaborate process models with several substrates must be utilized. Nevertheless, the very simple Monod kinetics describe the overall growth behaviour well and residuals are low.

In Fig 11, we have seen that the time differences in the exponential phases between replicates are well explained by the well-wise initial biomass concentrations $\vec{X}_0$. The choice of a hierarchical process models is further evaluated in Fig 15, which shows the estimated $\vec{X}_0$ with uncertainties for all replicates. For replicates with more evidence (longer time series), the posterior probability for their initial biomass concentration is concentrated in a narrow interval, whereas $X_0$ in wells with little evidence was estimated with considerably more uncertainty. The posterior for the group mean $X_{0,\mu}$ is concentrated at $\mathrm{HDI}^{90\ \%}_{X_{0,\mu}} = [0.251,\ 0.286]\ \frac{g}{L}$, close to the theoretical concentration $(0.25\ \frac{g}{L})$ expected from the experimental design.

Overall, the well-wise modeling of initial biomass concentrations as well as the separate modeling of replicates allowed us to account for inevitable differences between wells, while inferring the key process model parameters from all data. The combination of `calibr8` and `murefi` made it possible to construct a process models of our application example with little code and apply both optimization (MLE) and Bayesian inference (MCMC) without needing to

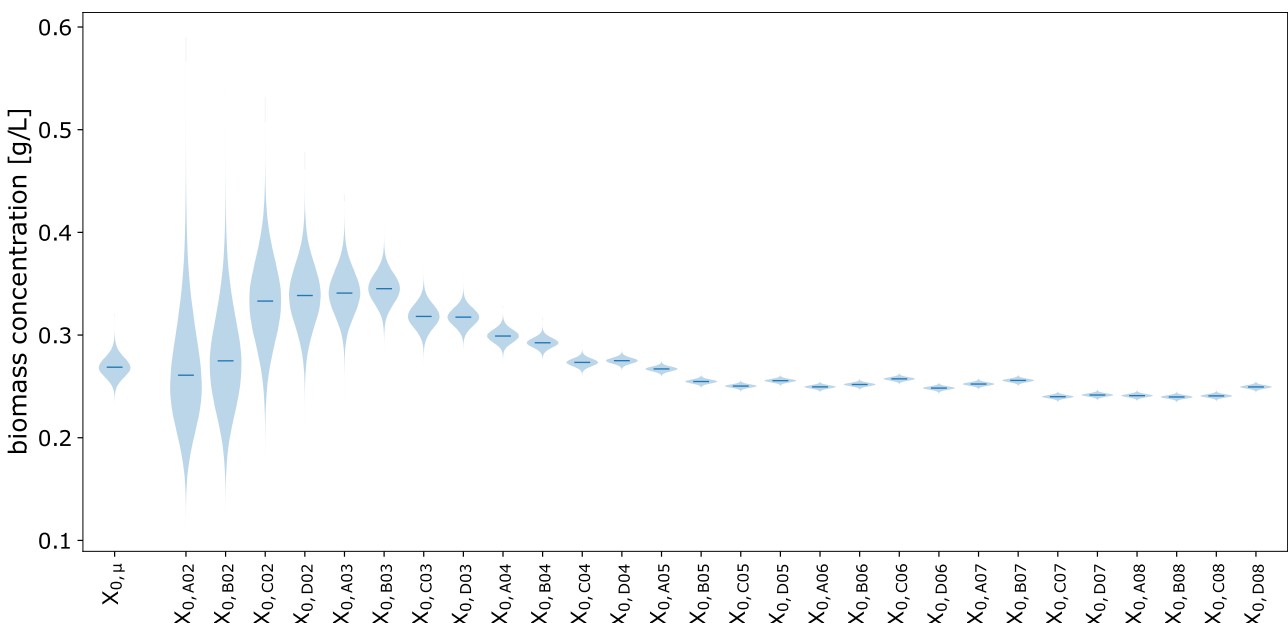

**Fig 15. Posterior group mean and well-specific initial biomass concentrations $X_0$.** Variability between the growth curves in separate wells is described by well-specific initial biomass concentrations $X_{0,\mathrm{well}}$. Their posterior probability distribution is wide if the well was sacrificed early (left) and narrows down with the number of observed time points (right). Their common hyperprior (a.k.a group mean prior) $X_{0,\mu}$ for the mean of each $X_{0,\mathrm{well}}$ was updated to a posterior with $\mathrm{HDI}^{90\ \%}_{X_{0,\mu}} = [0.250,\ 0.288]\ \frac{g}{L}$.

change any implementation details ([Box 4 Code 4](), [Box 5 Code 5]()). Our application example showed that Bayesian inference with ODE-based process models to 28 parallel cultures with hundreds of observations is not only technically feasible, but also accessible without deep understanding of probabilistic programming frameworks.

As implied in the famous quote by George E.P. Box—*"All models are wrong, but some are useful."*—also our Monod kinetics process model does not describe every aspect of the data, but is a powerful tool to quantify key process parameters under uncertainty. From its (in)accuracies, we can gain insight into the bioprocess and generate new hypotheses about the biological process or measurement system that are yet to be understood. In our case, the uncertainty quantification of process model parameters can become the cornerstone of bioprocess development by facilitating robust and intuitive statistical analysis or Bayesian decision-making.

## 4.3 Comparison with existing modeling software

A multitude of statistical software tools exist, many of which can be used for data analyses similar to the ones presented in this work. The technical complexity of performing such analyses, however, depends strongly on the technical capabilities of the software package. A comparison to relevant packages with similar scope and use-cases is given in [Table 2]().

For higher-throughput analyses and flexibility in the data analysis workflow, the user interface of statistical analysis software is particularly important. Most tools provide interfaces for popular scripting languages such as Python, R or MATLAB, but the model definition is in some cases delegated to a domain-specific programming language (DSL). For a broad application of calibration models, it is important that they are modular. Software like COPASI considers calibration only in the context of the ODE model and likelihoods cannot be customized. With modeling toolboxes such as Data2Dynamics or PESTO, custom calibration models and likelihoods can be realized, but they must be implemented manually as part of the objective function. This does not only require advanced expertise, but is also more error prone than working with a PPL directly.

pyPESTO, the unpublished Python implementation of PESTO, allows for customized noise models using the PETab data format [71] for model definition. However, PETab is currently limited to Normal- or Laplace-distributed noise [72]. Moreover, a straight-forward ODE

**Table 2. Comparison with related software packages.** DSL: Domain-Specific Language, GUI: Graphical User Interface.

| | User interfaces | Modularity of likelihood model | Required expertise | MCMC | ODE model construction | License |
|---|---|---|---|---|---|---|
| **murefi, calibr8** | Python | Modular | Low | Yes, with auto-diff | Templated | AGPLv3 |
| **PyMC** [53] | Python | Manual | Medium | Yes, with auto-diff | Manual | Apache 2.0 |
| **COPASI, PyCoTools3** [16, 17] | GUI, Python | No | Medium | No | Templated | Artistic 2.0, LGPL |
| **Data2Dynamics** [14] | MATLAB, DSL | Manual | Medium | Yes | Manual | Not specified |
| **PESTO** [15] | MATLAB | Manual | High | Yes | Manual | BSD-3 |
| **pyPESTO** | Python | Manual | High | Yes | SBML, PETab | BSD-3 |
| **Stan** [55] | DSL | Manual | High | Yes, with auto-diff | Manual | BSD-3 |
| **brms** [69] | R, Formula-based | Modular | Low | Yes, with auto-diff | N/A | GPLv2 |
| **JMP** [70] | GUI, HTTP (plugin) | No | Medium | No | N/A | Propri-etary |

implementation is not supported in pyPESTO at the time of publication. Instead, it requires the use of AMICI [73] via PETab or SBML or manual implementation up to the objective function. In contrast, `calibr8` separates calibration modeling entirely from the process modeling workflow, thereby becoming a valuable toolbox for calibration tasks even without process modeling.

`murefi` allows the practitioner to directly formulate ODE models without the need to manually provide the Hessian or derivative. For compatibility, a `murefi` objective, which returns a scalar tensor variable, might be used as an Aesara objective in pyPESTO to access its functionalities. Together with `calibr8`, this modular design allows to seamlessly use custom likelihood models in advanced ODE process models, a feature that we have not found with other software.

An important criterion for usability of calibration software is the required expertise. Packages that implement the foundations of model construction, auto-differentiation and definition of probability densities reduce the mathematical complexity and allow users with little technical expertise to perform advanced statistical analyses. `calibr8` and `murefi` are beginner-friendly, which is also evident from the simplicity of code examples [22, 49] compared to other tools [74, 75].

Bayesian analysis through MCMC methods is available through most modeling packages. Efficient, gradient-based state-of-the-art MCMC algorithms however are only readily available with probabilistic programming languages such as PyMC or Stan because they provide the necessary auto-differentiation of models. Finally, experimental replicates or hierarchical structures require replication and nesting of ODE models. Instead of manually expanding the differential equation system to match these requirements, templating approaches as they are used in `murefi` or COPASI can facilitate rapid model construction.

Concerning the accessibility for the systems biology research community, the recently published PETab data format, although currently limited to SBML models, is an interesting directive towards standardization of parameter estimation problems. However, PETab is a text file format specification and its functionalities are therefore limited to, for example, noise distributions that were included in the specification. Furthermore, it combines functionality that we deliberately split into two packages because we found calibration models to be much more generalizable. In the design of `calibr8` and `murefi` we chose to not specify a data file format, but in contrast implement modeling libraries that can be extended at run time, thus enabling practitioners to use custom noise distributions as shown in our documentation [22]. Another extension at runtime could be the export of a `calibr8` model to a PETab file, but the current PETab specification would constrain the calibration model to univariate inputs and untransformed, log, or log10 Normal or Laplace noise distributions. The separation between calibration and process models also enables greater modeling flexibility, as demonstrated by the hierarchical prior in Box 5 Code 5 or the hierarchical calibration example in the `calibr8` documentation [22].

## 5 Conclusions

In this paper, we introduced the general concept of calibration models and presented `calibr8`, an object-oriented Python toolbox that is applicable to both analytical calibration and inference of process models. Our open-source software allows to easily implement and analyze calibration models by providing a number of convenience functions, for example an asymmetric logistic function with an intuitive parametrization and a function to obtain the most important diagnostic plots in one line of code. It thus gives users without a background in statistics access to quantitative linear and non-linear calibration models, as well as Bayesian

uncertainty quantification. Furthermore, the implementation through a suite of extendable Python classes allows advanced modelers to customize the technique to a variety of applications. In comparison to existing software, the unique combination of modular likelihood functions from `calibr8` with objectives and (hierarchical) datasets from `murefi` enables a fully Bayesian, Pythonic approach to calibration and process modeling that could so far only be achieved by cumbersome manual implementation or combination of various libraries.

In our work, we demonstrated how the versatile asymmetric logistic calibration model can be applied to bioanalytical calibration tasks. Furthermore, we showed how combining the concept of calibration models with process models allows to gain process insight into a biological process. Especially in combination with `murefi`, our package to set up multi-replicate models, `calibr8` is suitable for high-throughput experimentation because of the flexible interface that allows to analyze data via optimization or MCMC. Uncertainty quantification is covered within the scope of the toolbox and enables easy identification of critical parameters. By making Bayesian inference of ODE models easy to implement, `calibr8` and `murefi` bridge the gap between bioprocess modeling and an entire portfolio of methods, such as Bayesian model comparison or decision-making.

Well-chosen calibration models eradicate the effect of systematic errors in measurements and allow the practitioner to focus a data analysis on the underlying process. In our application example, the non-linear biomass calibration model was required to identify lack-of-fit in the Monod model based on growth behaviour alone. We also identified the biomass overshoot at the beginning of the stationary phase as an interesting target for further investigation, e.g. by automated microscopy of cells during cultivation.

`calibr8` greatly reduces the workload of calibration tasks. For example, the systematic, model-based approach allows the user to quantify batch effects between calibration experiments; repetition of calibration measurements could thus be highly reduced. With `calibr8`, we provide a versatile toolbox that we believe to be beneficial not only for the biotechnology community, but for various calibration tasks in experimental sciences.

## Supporting information

**S1 Fig. Comparison of Normal vs. Student-*t* (log) probability density functions.** In the left chart, the probability density function (PDF) of a Normal distribution, as well as two Student-*t* distributions with varying degree of freedom (*v*) are shown. Both distributions are parametrized by a location parameter *μ* that is equal to the mean and mode of these distributions. In addition to *μ*, the Normal is parametrized by its standard deviation parameter *σ*, influencing the spread of the distribution. In contrast, the Student-*t* distribution has two spread parameters {scale, *v*} and is characterized by more probability mass in the tails of the PDF, not approaching 0 as quickly as the PDF of the Normal. With increasing *v*, the Student-*t* distribution becomes more similar to the Normal distribution. The log probability density (right) of the Normal distribution accelerates has a quadratic dependency on the distance to the mean, whereas the log-PDF of the Student-*t* distribution does not go to extreme values as quickly. Because of this property, the Student-*t* distribution causes less numerical problems at extreme values.
(PDF)

**S2 Fig. Pair plot of marginal posterior distribution.** Axis labels mark the 90% HDI and subplot axis limits are set at the 98% HDI. Units are $h^{-1}$ for $\mu_{\max}$, $\frac{g_{\text{glucose}}}{g_{\text{biomass}}}$ for $Y_{\text{XS}}$ and $\frac{g}{L}$ for $S_0$ and $X_0$.
(PDF)

**S1 File. Reparametrization of asymmetric logistic function.**
(PDF)

**S2 File. Mathematical notation of calibration models.**
(PDF)

**S3 File. Implementation, planning and saving of calibrations.**
(PDF)

**S4 File. Parameter mapping for fitting of Monod kinetics.**
(PDF)

**S1 Table. Parameter estimates from MLE and MCMC.** Maximum likelihood estimates, posterior sample means, standard deviation, HDI interval and $\hat{R}$ statistic.
(XLSX)

## Acknowledgments

First developments of data structures for multi-replicate modeling were made by Michael Osthege in the Theoretical Systems Biology group of Prof. Roland Eils at the German Cancer Research Center under the supervision of Dr. Stefan Kallenberger.

## Author Contributions

**Conceptualization:** Laura Marie Helleckes, Michael Osthege, Marco Oldiges.

**Data curation:** Laura Marie Helleckes, Michael Osthege.

**Formal analysis:** Laura Marie Helleckes, Michael Osthege.

**Funding acquisition:** Marco Oldiges.

**Investigation:** Laura Marie Helleckes, Michael Osthege.

**Methodology:** Laura Marie Helleckes, Michael Osthege.

**Project administration:** Marco Oldiges.

**Software:** Laura Marie Helleckes, Michael Osthege.

**Supervision:** Eric von Lieres, Marco Oldiges.

**Visualization:** Laura Marie Helleckes, Michael Osthege.

**Writing – original draft:** Laura Marie Helleckes, Michael Osthege.

**Writing – review & editing:** Laura Marie Helleckes, Michael Osthege, Wolfgang Wiechert, Eric von Lieres, Marco Oldiges.

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
