## [Decision Letter · Decision Letter 0]

13 Oct 2021

Dear Professor Oldiges,

Thank you very much for submitting your manuscript "Bayesian calibration, process modeling and uncertainty quantification in biotechnology" for consideration at PLOS Computational Biology.

As with all papers reviewed by the journal, your manuscript was reviewed by members of the editorial board and by several independent reviewers. In light of the reviews (below this email), we would like to invite the resubmission of a significantly-revised version that takes into account the reviewers' comments.

We cannot make any decision about publication until we have seen the revised manuscript and your response to the reviewers' comments. Your revised manuscript is also likely to be sent to reviewers for further evaluation.

Sincerely,

Dina Schneidman

Software Editor

PLOS Computational Biology

Reviewer's Responses to Questions

**Comments to the Authors:**

Reviewer #1: Helleckes et al describe a computational framework for the Bayesian analysis of calibration and process models. The framework is implemented in python, open-source, extensively documented implements code in a object-oriented manner. The software is tested using continues integration and coverage analysis. Overall I want to commend the authors for the quality of the presented software. The manuscript is well written and easy to understand, but may benefit from some restructuring that I will describe below. The authors evaluate the framework on multiple examples of biotechnological processes, including experimental data, where they analysis calibration models individually, as well as combinations of calibration and process models.

Major Comments

1) While most of the text is well written and easy to follow, there are some instances where it is difficult or impossible to understand the text without looking at the code or figures. Examples:

(i) Figure 3: Description of what was done is only available in figure legends, this also needs to be described in the text.

(ii) 4.1.1: It would helpful to have a mathematical description of the calibration model (provide formulas)

(iii) Figure 10: It is hard to understand what was actually done and the reader has to guess based on the figure. Describe what was done first, then the results.

2) The (unpublished) python implementation of PESTO, pyPESTO https://github.com/ICB-DCM/pyPESTO), implements a similar feature set as the murefi/calibr8 including providing an interface to pyMC3 for Bayesian analysis. Still pyPESTO implements less modularity between process model and calibration model to enable use of adjoint sensitivity analysis. Similarly, the recently proposed PEtab format (https://journals.plos.org/ploscompbiol/article?id=10.1371/journal.pcbi.1008646) permits a similarly flexible specification of calibration models and multi-experiment process models (templating) in COPASI/data2dynamics/pyPESTO. Accordingly, I think it would be important to

(i) explicitely describe differences to what is possible with PEtab

(ii) demonstrate the greater flexibility by the modularity between calibr8 and murefi.

Regarding (i), I would also encourage the authors to implement support for the PEtab format and potentially propose an extension of the format based on the identified differences (I don't think these points are necessary for the scope of this review though)

Regarding (ii) I think the use of a hyperprior in Section 4.2.3 is an interesting example, but I think it would be more convincing if the nesting would be in the calibration model component and if the authors could demonstrate that joint analysis of process+calibration model is important to obtain accurate credibility intervals for parameters (by comparing stepwise sequential analysis, using synthetic data if necessary)

Am I correct in my understanding that the toolbox cannot be applied to mixed effect modeling? If I am mistaken, this would be quite convincing and a demonstration could replace point (ii).

A third way of addressing (ii) would be to demonstrate that the combination of calibr8/murefi/sunode enables the use of adjoint sensitivities (see https://journals.plos.org/ploscompbiol/article?id=10.1371/journal.pcbi.1005331) with complex noise models, but I expect that this would be a lot of work

3) The description in 3.2.6 seems to imply that murefi supports sensitivity analysis which would be necessary for the computation of the posterior gradient. Looking at the code, this does not seem to be supported though, but theoretically possible with sunode. It would be helpful if the authors could more extensively describe the support of gradient computation in calibr8/murefi.

Minor Comments

a) I believe that the abstract should mention that the authors apply their methods to ODE models.

b) l244, the text just mentions calibr8, which is only introduced later in the text

c) I have a hard time understanding the issue that is supposed to be illustrated by figure 3C, could the authors describe the issue in more detail, is there some statistical test to show the issue (and is such an analysis implemented in calibr8)?

d) I find it hard to believe that gradient based optimizers did not perform well for section 3.2.5. Did the authors try using the least squares algorithm and used a logarithmic transformation of process parameters?

e) the authors may want to cite https://doi.org/10.1093/bioinformatics/btw703 in section 2.4

f) I think "Frequentist" is more commonly used than "likelihoodist" (assuming that's what the authors wanted to say).

Reviewer #2: This proposal is focused on the description of a Python package for the use of Bayesian theory for the quantification of uncertainty from high-throughput cultivation experiments. The paper is technically sound, well written and all the codes are available on a Git server. Instructions about the installation on a Python idle are clearly given.This proposal has been submitted as a “software” article. As such, the software must be either widely used within the scientific community or have the promise of wide adoption by a broad community of users. However, as it is actually written, the target audience is quite limited. Specific comments are appended below.

- The background needs to be more precisely defined. The field of expertise of the authors are “bioprocess engineering”. However, as stated in the introduction, the software could be used by any researchers handling large set of microbial (or even cell culture) kinetics data, extending the target audience to the broad community if researchers involved in fields like systems and synthetic biology. Some part of the text should then be reformulated according to this comment.

- The term “microbioreactor” is used at several stages of the manuscript. This term is actually misleading, because the authors are used a commercial minibioreactor platform based on the use of deepwell. For the readers that are not familiar with the technology, the use of the term “micro” could give the feeling that the authors are using microfluidic cultivation device. This comment is also related to the above-mentioned in the sense that the technical terms in the text should be reformulated to target a broader audience of potential users (other example: I can imagine that only a limited number of specialists know what is a flowerplate).

- At this stage, the applicability of the software is limited to a cultivation device commercialized under the name of “Biolector” and involving the use of 48-well deepwell plates. Additionally, the Biolector can be coupled to a robotic liquid handling platform for off-line sampling. The authors have to discuss the potential extension of their software to other microplate-based devices.

- Bayes theory is typically easy to explain, but it is not the case of this paper. Section 2.1 should be expanded by considering an example.

- About the applications, only two are given in the manuscript. Another, very important, application is the measurement of the activity of gene circuits based on fluorescent reporters. For this application, it is also important to relate to biomass in order to obtain specific values. How could this application integrated in the package ?

- ODE models can be integrated in the calibration procedure. Is there any limit about the number of equations/parameters that can be handled ?

Reviewer #3: In the manuscript "Bayesian calibration, process modeling and uncertainty quantification in biotechnology", Laura Marie Helleckes, Michael Osthege, and coworkers present two Python software packages, calibr8 and murefi, for enabling more reproducible and automated calibration of mathematical models in biotechnology. The authors show the capabilities of these packages on several examples with real datasets collected by themselves.

The topic of parameter inference for mathematical models of biological systems have become a very relevant topic in the recent years. This can be achieved by a frequentist perspective, finding the maximum likelihood estimate using various optimization techniques, or by a Bayesian approach, which relies on the Bayes theorem and often carried through Markov chain Monte Carlo sampling.

One of the key problems is that frequently “handmade” computational pipelines are built for each specific problem which compromises their reproducibility and reusability. Efforts in the community to build automated pipelines exist, however these can be complex and require high expertise. In this manuscript, the authors tackle this problem by introducing two new software tools for model construction and calibration, which substantially ease the process and facilitate the usage to non-experts, in particular, focusing - but not limited to - biotechnology applications. Likewise, the key contributions is the development of two reusable open-source toolboxes/libraries. Overall, the manuscript is well written. I appreciate that the authors provide exemplary notebooks/code in the respective toolboxes, but I have to admit that I did not have the time to test them.

=====

Major

=====

As far as I can see, the only available loglikelihood implemented in calibr8 assumes a Student’s t distribution. This is a bit contradictory with the statement the authors make in L.67-69 regarding the need of having more flexible frameworks. Maybe the authors could include in their toolbox additional distribution assumptions such as Laplace, Gaussian and/or log-normal, to facilitate the users this flexibility. Moreover, I do not see how the user could easily implement a different custom noise model, in case this is possible it would be great to add a tutorial.

A very well-known standard format to encode ODE models is the SBML format (https://doi.org/10.1093/bioinformatics/btg015) which is supported by many existing toolboxes for modeling and parameter estimation. Including support to this in the toolboxes here presented would substantially increase the public and potential new users to the tool. I deeply encourage the authors to add support to SBML models although not only ODE models are used within these toolboxes.

Following on the line of my previous comment: Is there a model validator? What I mean is whether is there some sort of sanity checks for user defined models in calibr8 and murefi., e.g. positivity of the modeled species. This would make even better the user experience, since then, even the level of expertise required could be lowered. I am wondering if the authors thought of this option, and whether it would be possible to include it (or at least comment on this).

- L.88: “examples are Data2Dynamics [14] or PESTO [15], ... However, both tools are implemented in MATLAB and are thus incompatible with data analysis workflows that leverage the rich ecosystem of scientific Python libraries.” I would like to make the authors aware that I could find that the toolbox PESTO has been translated into Python, going under the name of pyPESTO (https://github.com/ICB-DCM/pyPESTO). I would encourage the authors to include it into their manuscript since it has been released since January 2019 and, therefore, adapt the comparison within toolboxes.

L.177: The statement is correct. However, parameter uncertainties can also be quantified from a frequentist perspective using optimization by the so-called method profile likelihoods (see https://doi.org/10.1093/bioinformatics/btp358). I am not aware whether this is known in the field of biotechnology, but definitely something to mention in a manuscript regarding parameter estimation and uncertainty quantification. In case this is not frequently used in this field, it could be also a novelty to add in the study (although not necessary).

L.205: Could some citation be added to this statement?

L.226-227: Could some citation be added to this statement?

L.244: Please reformulate the sentence, I could not understand which restriction is meant here.

=====

Minor

=====

L.112: Without losing generality, the authors could list some specific examples of other research fields.

Figure 1: I suggest to use the same label and line style in the two subplots for the Normal case.

L.142: “From a known list of parameters …” The word “known” here could lead to misunderstanding since the parameters may be actually unknown and need to be estimated. But I understand that when simulating they are actually “known”. Maybe this sentence could be rephrased (in case of finding a better formulation).

L.158: Please clarify that each individual pair y_{obs} and y_{pred} are the same length.

Figure 3: Please indicate in the figure as legend what the blue dots and green lines are. Having this on top of the caption description will facilitate the understanding of what is depicted. Same for the black dashed line.

Figure 7: Change the color scheme → colorblind proof

L.540: To the list of known samplers, please add “emcee” which is also a very popular python sampler (https://doi.org/10.1086/670067).

Figure 12: Please increase the separation between A and B. This helps to identify the right Y axis in A.

General: Revise the text for typos.

General: Please use consistent font sizes in the figures, for some, they are really small.

**Have the authors made all data and (if applicable) computational code underlying the findings in their manuscript fully available?**

Reviewer #1: Yes

Reviewer #2: Yes

Reviewer #3: Yes

PLOS authors have the option to publish the peer review history of their article (what does this mean?). If published, this will include your full peer review and any attached files.

Reviewer #1: No

Reviewer #2: **Yes: **Frank Delvigne

Reviewer #3: **Yes: **Elba Raimundez
---

## [Decision Letter · Decision Letter 1]

16 Jan 2022

Dear Professor Oldiges,

We are pleased to inform you that your manuscript 'Bayesian calibration, process modeling and uncertainty quantification in biotechnology' has been provisionally accepted for publication in PLOS Computational Biology.

Before your manuscript can be formally accepted you will need to complete some formatting changes, which you will receive in a follow up email. A member of our team will be in touch with a set of requests. Also please address reviewer 3 comments in this submission.

Best regards,

Dina Schneidman

Software Editor

PLOS Computational Biology

Reviewer's Responses to Questions

**Comments to the Authors:**

Reviewer #1: The authors have adequately addressed my concerns.

Still, the authors may want to consider using qq-plots to visualize differences in data/theoretical quantiles/percentiles in figure 3.

Reviewer #3: The authors have addressed the reviewer comments. The manuscript is improved. I reviewed this manuscript a second time with a closer attention to details within the text, from which I have only few minor comments.

L.17: Specify the name of the packages as done in the author summary

L.22 and L.24: the abbreviations can be introduced in the main body instead of in the abstract

L.82: “many methods assume” This is very generic, maybe list a few examples of such methods.

L.137: check for consistency style with or without italics for “Normal”, same for “Student-t” (and other words which I may have missed)

L.124: introduce here abbreviation for MLE

L.175: List some examples

L.185: previous chapter? Or previous subsection? Maybe avoid unclear references and point directly to the desired Section, e.g. “introduced in SectionX.X”

L.186: abbreviation already introduced

L.188: e.g. here Normal is without italics (check for style consistency in the whole text), same for “Bayesian inference” (see L.193 and L.124)

L.213: indeed the choice is arbitrary, however 99%, 95% and 90% are predominantly used maybe this information can be added

Figure3: Likelihood bands reported in the caption differ from those in the figure legend (A), please check which are the correct ones.

General: check for writing style of specific words in italics vs non-italics

L.301: ODE is already introduced as abbreviation

L.566: which prior assumptions?

**Have the authors made all data and (if applicable) computational code underlying the findings in their manuscript fully available?**

Reviewer #1: None

Reviewer #3: Yes

PLOS authors have the option to publish the peer review history of their article (what does this mean?). If published, this will include your full peer review and any attached files.

Reviewer #1: No

Reviewer #3: No

---

## [Editor Report · Acceptance letter]

7 Feb 2022

PCOMPBIOL-D-21-01126R1

Bayesian calibration, process modeling and uncertainty quantification in biotechnology

Dear Dr Oldiges,

I am pleased to inform you that your manuscript has been formally accepted for publication in PLOS Computational Biology. Your manuscript is now with our production department and you will be notified of the publication date in due course.

With kind regards,

Livia Horvath
